# GML-NeRF: Gate-guided Mutual Learning Framework for Neural Rendering

## Abstract

Although the neural radiance field (NeRF) exhibits high-fidelity visualization on the rendering task, it still suffers from rendering defects in complex scenes. One of the primary reasons is the limited model capacity. However, directly increasing the network's width and depth cannot significantly improve the rendering quality. To address this issue, existing work adopts scene partitioning and assigns different 3D points to different network parameters. However, a 3D point may be invisible to some rays due to occlusions in complex scenes. On such a point, training with those rays that do not contain valid information about the point might interfere with the NeRF training. Based on the above intuition, we allocate model parameters in the ray dimension and propose a **G**ate-guided **M**utual **L**earning framework for neural rendering **(GML-NeRF)**. Specifically, we construct an ensemble of sub-NeRFs and train a soft gate module to assign the gating scores to these sub-NeRFs based on specific rays. The gate module is jointly optimized with the sub-NeRF ensemble, enabling it to learn the preference of sub-NeRFs for different rays automatically. Furthermore, we introduce depth-based mutual learning to enhance the rendering consistency among multiple sub-NeRFs and mitigate the depth ambiguity. Experiments on five diverse datasets demonstrate that GML-NeRF can enhance the rendering performance across a wide range of scene types compared with existing single-NeRF and multi-NeRF methods.

## 1 Introduction

Novel view synthesis is an important task within the domains of computer vision and computer graphics, playing an essential role in a variety of applications, such as autonomous driving, augmented reality, virtual reality, game rendering, and so on. Recently, Neural Radiance Field (NeRF) (Mildenhall et al., 2021) has emerged as a promising solution, achieving high-fidelity visualizations on the novel view synthesis task. It implicitly encodes 3D scenes through neural networks and trains the networks using multi-view consistency and volume rendering.

Despite NeRF's excellent scene representation ability, it still suffers from rendering defects when dealing with complex scenes, such as 360-degree unbounded scenes (Zhang et al., 2020; Barron et al., 2022) and large indoor or outdoor scenes with free shooting trajectories (Wang et al., 2023; Turki et al., 2022; Tancik et al., 2022). One of the main reasons is the limited capacity of the NeRF model. However, directly increasing the network's width and depth yields marginal improvement in the rendering quality (Müller et al., 2022).

To address the limitation, a recent scene partitioning method proposes to allocate different 3D points to multiple NeRFs and trains them independently (Zhenxing & Xu, 2022). This approach, where each NeRF's parameters encode specific regions of the 3D space, indeed leads to improved rendering performance compared to a single NeRF model. Nonetheless, this approach might struggle to cope with complex scenes (with many occlusions and arbitrary shooting trajectories).

Let us consider a simple case of a 360-degree unbounded scene with a central object (truck) and a distant object (car), in which NeRF needs to encode both objects. As illustrated in Figure 1a, a 3D point located on the distant object can be observed from ray-1 and ray-2, but is invisible to ray-3 due to the occlusion presented by the central object. If we use the point-based scene partitioning scheme that assigns 3D points around the two objects to two sub-NeRFs respectively, when NeRF-1 learns the 3D point located on the distant object, all three rays will be used for training. However,

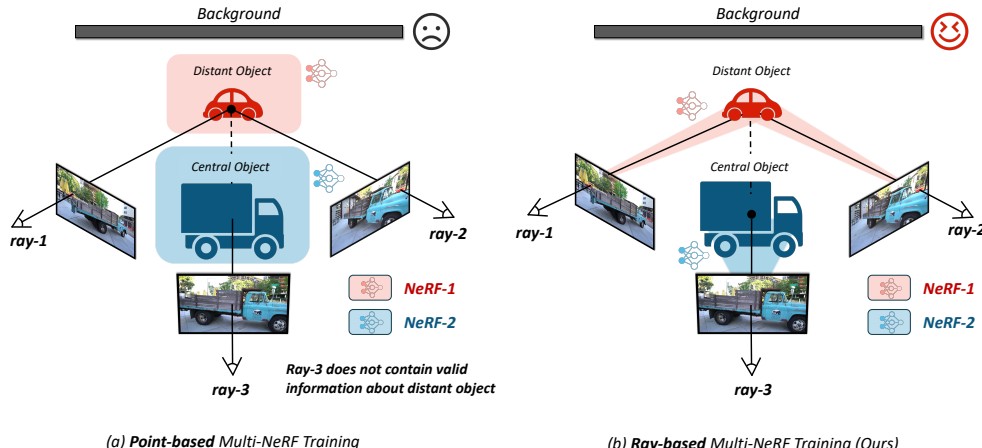

Figure 1: A case in 360-degree unbounded scenes (bird-eye view). (a) In the point-based multi-NeRF framework, model parameters are allocated in the point dimension, which is not visibility-aware. Due to the occlusion by central object, distant object is invisible to ray-3 and training NeRF-1 with ray-3 could interfere with the rendering. (b) The ray-based multi-NeRF framework considers the variable visibility of objects to different rays and allocates parameters in the ray dimension.

ray-3 does not contribute any meaningful information about the distant object, potentially interfering with the training process of NeRF-1. In contrast, considering the different visibility of the object to different rays, our intuition is that assigning the rays terminating at the distant object to NeRF-1 and the rays terminating at the central object to NeRF-2 could be better, as shown in Figure 1b.

The above intuition inspires us to **allocate model parameters in the ray dimension** rather than the point dimension, realizing a visibility-aware multi-NeRF framework. To this end, we propose a gate-guided multi-NeRF mutual learning framework for neural rendering (GML-NeRF). Within the GML-NeRF framework, an ensemble of sub-NeRFs assigns different model parameters to different rays through a soft gate module. With the help of the soft gate module, sub-NeRFs' outputs are fused by post-volume-rendering fusion to yield final rendering results. Notably, the gate module is jointly optimized with NeRF, allowing it to automatically learn the preference of each sub-NeRF for various rays in an end-to-end manner. This **learnable parameter allocation** design makes GML-NeRF generally applicable to diverse scenes, which stands in contrast to prior multi-NeRF methods (Turki et al., 2022; Tancik et al., 2022) that rely on manually defined allocation rules.

Building upon the gate-guided multi-NeRF framework, we introduce a depth-based mutual learning method to enhance the rendering consistency among multiple sub-NeRFs. Specifically, in addition to learning the ground-truth colors, sub-NeRFs teach each other with their rendered depth. Traditional NeRF methods may struggle with generalization to novel views despite accurately rendering training views, as they often fail to capture precise geometry (Deng et al., 2022; Zhang et al., 2020). In contrast, our depth-based mutual learning approach serves as a form of geometric regularization, alleviating the depth ambiguity and thereby avoiding overfitting.

To verify the effectiveness and general applicability of GML-NeRF, we conduct experiments on various datasets, including real object dataset, 360-degree unbounded scenes, and large outdoor/indoor scenes with free shooting trajectories. The results show that GML-NeRF effectively boosts rendering quality and outperforms other multi-NeRF methods across different types of scenes.

## 2 RELATED WORK

### 2.1 NEURAL RADIANCE FIELD

Neural Radiance Field (NeRF) (Mildenhall et al., 2021) has received much attention since it was proposed. It uses MLPs to implicitly represent 3D objects or scenes, achieving realistic rendering results. Due to the success of NeRF in high-quality rendering, there have been intensive studies on NeRF's extension, including but not limited to increasing NeRF's training or inference efficiency (Yu

et al., 2021; Fridovich-Keil et al., 2022; Reiser et al., 2021; Sun et al., 2022; Müller et al., 2022), applying NeRF to complex scenes (large/unbounded/poor-textured) (Martin-Brualla et al., 2021; Zhang et al., 2020; Barron et al., 2022; Wei et al., 2021; Tancik et al., 2022; Turki et al., 2022; Zhenxing & Xu, 2022), applying NeRF to other tasks (surface reconstruction/scene editing) (Yariv et al., 2021; Oechsle et al., 2021; Wang et al., 2021a; Liu et al., 2021; Yang et al., 2021; 2022), increasing NeRF rendering quality in few-shot setting (Jain et al., 2021; Kim et al., 2022; Niemeyer et al., 2022; Deng et al., 2022). In this work, we aim to increase NeRF's rendering quality in complex scenes, and our GML-NeRF acts as a multi-NeRF framework, which can leverage the techniques proposed by these single-NeRF researches.

## 2.2 MULTI-NERF REPRESENTATION

Compared to rendering 3D objects and face-forward small scenes in the original NeRF (Mildenhall et al., 2021), some work applies NeRF to the more complex scenes. Due to the limited model capacity, the multi-NeRF method is widely adopted to scale up model capacity and allocate model parameters to improve the rendering quality. For outdoor unbounded scenes, NeRF++ (Zhang et al., 2020) proposes the sphere inversion transformation to map an infinite space to a bounded sphere first, and it uses two NeRFs to model the foreground and background regions respectively. For large scenes, Block-NeRF (Tancik et al., 2022), Mega-NeRF (Turki et al., 2022), and Switch-NeRF (Zhenxing & Xu, 2022) partition the scenes into multiple parts which are processed by different NeRFs, in image, pixel and point dimension respectively. The former two methods adopt hand-crafted scene partitioning schemes based on manually defined data allocation rules, which are impractical and have limited application scenes. Besides, they adopt a hard allocation scheme and need other processes to improve rendering consistency. Switch-NeRF (Zhenxing & Xu, 2022) implements a learning-based scene partition scheme motivated by Mixture-of-Experts (MoE) (Shazeer et al., 2017a). Despite its effectiveness in drone scenes, it allocates model parameters in the point dimension, which limits the rendering performance in the more complex scenes with occlusions. In this work, we propose a gate-guided multi-NeRF mutual learning framework, performing the allocation in the ray dimension. Compared to other multi-NeRF methods, GML-NeRF boosts the rendering quality in various types of scenes without the need for prior scene knowledge.

## 3 PRELIMINARY

NeRF (Mildenhall et al., 2021) uses neural networks to represent 3D scenes implicitly. Two MLPs model the density and color of spatial points respectively. The input of density MLP $F_\sigma$ is the 3D coordinate of spatial point $\mathbf{x}$. As the contribution of each spatial point to the rendered color can be different when observing from different views, the input of color MLP $F_c$ includes view direction $\theta$ and the feature $f$ output by density MLP. NeRF proposes the volume rendering method to render each pixel of an image. Specifically, each pixel on the images corresponds to a ray. NeRF samples $N$ points along the ray and renders the pixel's color $\hat{C}(\mathbf{r})$ by discretely summing density $\sigma_i$ and color $\mathbf{c}_i$ of each point $i$, which approximates the integral $C(\mathbf{r})$ as follows:

$$C(\mathbf{r}) = \int_0^{+\infty} w(t)\mathbf{c}(t)dt \quad \hat{C}(\mathbf{r}) = \sum_{i=1}^{N} w_i \mathbf{c}_i, \tag{1}$$

$$T_i = \exp\left(-\sum_{j=1}^{i-1} \sigma_j \delta_j\right) \quad w_i = T_i\left(1 - e^{-\delta_i \sigma_i}\right), \tag{2}$$

where $t_i$ is the distance between $i$-th sample's position and the starting point of the ray, $\delta_i = t_{i+1} - t_i$ is the distance between adjacent samples and $T_i$ represents the probability that the ray travels from the start to point $i$ without hitting. The NeRF optimization is based on color supervision.

## 4 GML-NERF

NeRF faces the challenge of limited model capacity when rendering complex scenes (Zhang et al., 2020; Wang et al., 2023; Zhenxing & Xu, 2022). However, directly increasing the number of model

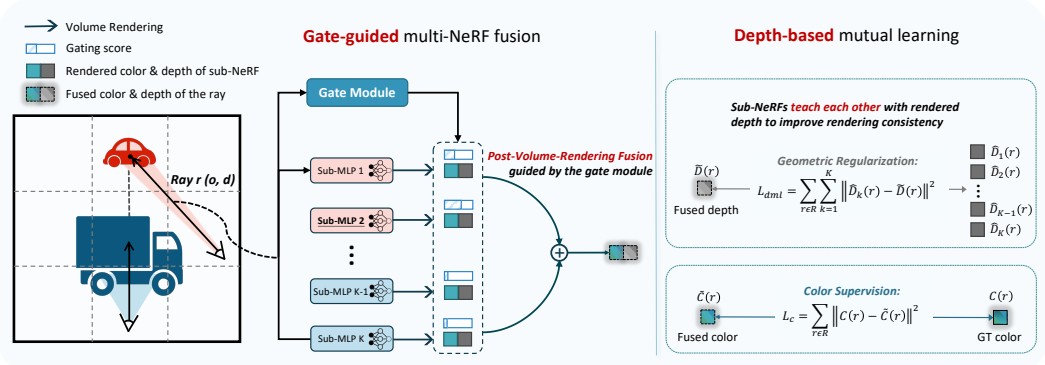

Figure 2: The overview of GML-NeRF. We construct a multi-NeRF framework based on the hybrid representation, where the feature grid is shared for all sub-NeRFs and the MLP decoders are independent. **(Left)** Given a ray, the soft gate module encodes the ray's data and outputs a soft score. Then, guided by the gating score, sub-NeRFs' outputs are fused after the volume rendering process. **(Right)** The fused rendered depth of the ray is used to regularize each sub-NeRF's geometric encoding and increase the rendering consistency.

parameters yields marginal improvement in the rendering quality (Müller et al., 2022), posing an important research question: "how to effectively scale up the capacity of NeRF". While the multi-NeRF methods have been proposed as an effective technique in response to this question, they still faces limitations in handling complex scenes (with many occlusions and arbitrary shooting trajectories). In this work, we propose a gate-guided mutual learning framework (GML-NeRF), effectively scaling up the model's capacity to handle complex scenes by allocating model parameters in the ray dimension. Figure 2 gives an overview of GML-NeRF.

In the following, we first analyze the limitations of existing multi-NeRF methods and demonstrate the motivations of GML-NeRF in Section 4.1. Then, we introduce the design of gate-guided multi-NeRF fusion in Section 4.2 and depth-based mutual learning method in Section 4.3. Finally, we describe the training losses of GML-NeRF in Section 4.4.

## 4.1 MOTIVATIONS OF GML-NERF

Existing multi-NeRF methods can be categorized into two categories according to the dimension of model parameter allocations: point- and ray-based multi-NeRF methods.

**Point-based multi-NeRF method.** These methods divide the 3D space in the point dimension (Zhang et al., 2020; Wang et al., 2023; Zhenxing & Xu, 2022). 3D points in different regions are computed by different sub-NeRFs. Although they perform well on open scenes with few occlusions, *they are not "visibility-aware", which might hinder their performances on complex scenes with many occlusions*: These methods do not consider the different visibility of a target region to different views. For example, the front side of an object is not visible when it is observed from the back view or it is blocked by an occlusion. Training the sub-NeRF with rays that do not contain any valid information about the target region might interfere with the training.

**Ray-based multi-NeRF method.** Existing ray-based multi-NeRF methods allocate the training rays to different sub-NeRFs and train sub-NeRFs independently (Tancik et al., 2022; Turki et al., 2022). Block-NeRF and Mega-NeRF perform the ray allocation in the image-granularity and pixel-granularity respectively. Both of them need a manually defined allocation rule, which requires concrete knowledge of the application scenes and cannot be easily adapted to other types of scenes. For example, the former work trains sub-NeRFs in large-scale road scenes with prior knowledge of the image shooting position distribution on the road, and the latter one trains sub-NeRFs in the open drone scenes and allocates the rays by partitioning the intersecting positions between the rays and a horizontal plane. Despite their promising performances on these specific types of scenes, *defining a*

*ray allocation rule for complex scenes lacking prior scene-specific knowledge remains challenging, especially when the scenes are captured with arbitrary shooting trajectories.*

**Requirements on the multi-NeRF framework.** As discussed above, the complex scenes may contain many occlusions and be captured with arbitrary shooting trajectories, which poses challenges for the point-based multi-NeRF methods and manually defined ray allocation rules. For these complex scenes, an improved method should be both **visibility-aware** and **easy to apply** to different types of scenes, motivating us to design a ray-based multi-NeRF framework in a learnable way.

## 4.2 GATE-GUIDED MULTI-NERF FUSION

In order to allocate model parameters in the ray dimension, we design a multi-NeRF model structure and introduce a soft gate module to produce soft NeRF allocation for each ray in a learnable way.

**Multi-NeRF Structure.** Considering the importance of training efficiency on the practicability of NeRF frameworks, we combine the implicit and explicit representations following Instant-NGP (Müller et al., 2022). When performing a forward computation with a single NeRF, we first extract the feature of the 3D point from a feature grid, and then we compute its density and color using tiny MLP networks. Despite the training efficiency of the hybrid representation, it needs to explicitly store most of the learnable parameters, leading to increasing memory overhead. To mitigate this issue, we employ a shared feature grid among sub-NeRFs and keeping MLP decoders independent, avoiding a significant increase in the number of parameters and training complexity. Besides, as different rays may pass through the same region of 3D space, weight sharing for the feature grid might help training, while ray specialties are still maintained by independent MLP decoders.

**Soft Gate Module.** We incorporate a soft gate module to assign gating scores to the sub-NeRFs for each ray. The soft gate module is jointly optimized with NeRF, enabling it to learn the preference of each sub-NeRF for different rays in an end-to-end manner. In contrast to manually assigning training rays to sub-NeRFs, this learnable parameter allocation design makes GML-NeRF **generally applicable to diverse scenes lacking prior scene-specific knowledge**, as we will verified in Section 5.2. In Section 5.2, we will also show that **the gate module can learn to assign reasonable gating scores that reflects the object visibility of rays**, aligning with our intuition that visibility-aware parameter allocation is important.

Specifically, we employ a four-layer MLP followed by a Softmax as the gate module. The soft gate module takes the starting point and direction $(\boldsymbol{o},\boldsymbol{d})$ of a ray $\mathbf{r}$ as the input, and outputs the soft gating scores $\boldsymbol{G}(\mathbf{r})$ of the multiple sub-NeRFs associated with this ray. Instead of applying any sparsification strategies on the gating score $\boldsymbol{G}(\mathbf{r})$ as in previous work (Zhenxing & Xu, 2022), such as top-k gating function (Shazeer et al., 2017b), we use soft gating scores to enhance the consistency of the rendered results.

As discussed in Section 3, each ray corresponds to a pixel on the image. Following the volume rendering process, we can obtain $K$ rendered colors for each ray, where $K$ is the number of sub-NeRFs. Subsequently, multi-NeRFs' outputs are fused in a post-volume-rendering ordering to obtain the final rendering results. The fused color $\tilde{C}(\mathbf{r})$ of the ray $\mathbf{r}$ can be written as below:

$$\tilde{C}(\mathbf{r}) = \sum_{k=1}^{K} G_k(\mathbf{r})\hat{C}_k(\mathbf{r}),  \tag{3}$$

where $G_k(\mathbf{r})$ is the $k$-th element of gating score $\boldsymbol{G}(\mathbf{r})$ and $\hat{C}_k(\mathbf{r})$ is the rendered color of $k$-th sub-NeRF for the ray $\mathbf{r}$.

## 4.3 DEPTH-BASED MUTUAL LEARNING

Within our multi-NeRF framework, we introduce a mutual learning method to enhance the rendering consistency of sub-NeRFs, wherein each sub-NeRF not only learns from ground truth but also learns from each other. Due to the lack of the ground-truth for per-ray depth, NeRF may fail to learn accurate geometry despite accurately rendering training views, which adversely affects its generalization to novel views. To address this, we perform mutual learning with the rendered depths of sub-NeRFs, which serves as a form of geometric regularization. The per-ray depth estimation $\hat{D}(\mathbf{r})$

can be written as Equation 4, where $t_i$ is the i-th sample's distance from the starting point on the ray.

$$\hat{D}(\mathbf{r}) = \sum_{i=1}^{N} w_i t_i, \tag{4}$$

In practice, we first fuse the rendered depths of sub-NeRFs guided by the gating score $\boldsymbol{G}(\mathbf{r})$, and then we use the L2 distance to quantify the match of each sub-NeRF's rendered depth $\hat{D}_k(\mathbf{r})$ and the fused depth $\tilde{D}(\mathbf{r})$. Our depth-based mutual learning loss is defined as below, where $\mathcal{R}$ is the set of sampled rays:

$$L_{dml} = \sum_{\mathbf{r} \in \mathcal{R}} \sum_{k=1}^{n} \|\hat{D}_k(\mathbf{r}) - \tilde{D}(\mathbf{r})\|^2, \tag{5}$$

Compared to directly averaging sub-NeRFs' depth predictions as in traditional mutual learning frameworks, the gate-guided fused depth $\tilde{D}(\mathbf{r})$ is more accurate, as the gating score $\boldsymbol{G}(\mathbf{r})$ reflects the prediction confidence of each sub-NeRF for the ray $\mathbf{r}$.

### 4.4 THE OVERALL TRAINING LOSS

The overall loss function of GML-NeRF is given by:

$$L = L_c + \lambda_1 L_{dml} + \lambda_2 L_{cv}, \tag{6}$$

where $L_c = \sum_{\mathbf{r} \in \mathcal{R}} \|C(\mathbf{r}) - \tilde{C}(\mathbf{r})\|^2$ ($C(\mathbf{r})$ is the ground truth color value of ray $\mathbf{r}$) is the rendering loss. And $\lambda_1$ and $\lambda_2$ are the weights for regularization terms. $L_{cv}$ is the balancing regularization on the Coefficient of Variation of the soft gating scores, which encourages a balanced allocation of model parameters for training rays and prevents the gate module from collapsing onto a specific sub-NeRF. The details of $L_{cv}$ are described in Appendix A.1.

## 5 EXPERIMENTS

### 5.1 DATASETS AND BASELINES

**Datasets.** We use five datasets from different types of scenes to evaluate our GML-NeRF. (1) Object dataset: we take *Masked Tanks-And-Temples dataset (MaskTAT)* (Knapitsch et al., 2017) for evaluation, which are photographed objects with masked background; (2) 360-degree inward-facing datasets: we take *NeRF-360-v2 dataset* (Barron et al., 2022) and *Tanks-And-Temples (TAT) dataset* with unmasked background (Knapitsch et al., 2017) to evaluate on scenes with large dynamic depth range; (3) free shooting-trajectory datasets: we conduct experiments on *Free-Dataset (Wang et al., 2023)* and *ScanNet dataset* (Dai et al., 2017), which are large outdoor and indoor scenes respectively. Both larger view range and more irregular shooting trajectories pose greater challenges for NeRF rendering.

**Baselines.** We compare our GML-NeRF with two types of methods: one type uses the grid-based NeRF framework as we do, including PlenOctrees (Yu et al., 2021), DVGO (Sun et al., 2022), Instant-NGP (Müller et al., 2022) and F2-NeRF (Wang et al., 2023). The other one is the MLP-based NeRF method, including NeRF (Mildenhall et al., 2021), NeRF++ (Zhang et al., 2020), mip-NeRF (Barron et al., 2021) and mipNeRF360 (Barron et al., 2022), which is inefficient in training and needs almost one day for training in complex scenes. Note that we also implement the NGP-version of Block-NeRF (Tancik et al., 2022), Switch-NeRF (Zhenxing & Xu, 2022) and Mega-NeRF (Turki et al., 2022) to validate the superiority of GML-NeRF to other multi-NeRF methods. See Appendix A for the implementation details and Appendix E for the discussion of Mega-NGP.

### 5.2 COMPARATIVE STUDIES

**GML-NeRF achieves higher rendering quality than existing single- and multi-NeRF methods.** We report the main quantitative results on the complex scenes and the object dataset in Table 1 and Table 6 respectively. Within no more than one hour of training, GML-NeRF achieves higher rendering quality compared to other fast training methods and multi-NeRF methods, including Switch-NGP and Block-NGP. We can also see that while GML-NeRF is designed for complex scene rendering, it can also improve the rendering performance of objects. We also integrate GML-NeRF with

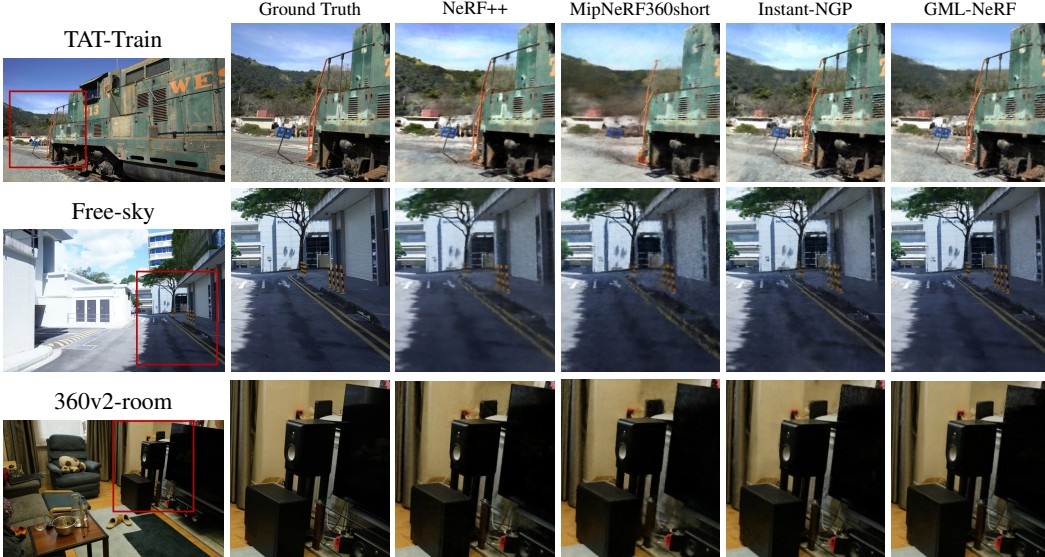

Figure 3: Qualitative comparisons on three complex scenes. GML-NeRF achieves better recovery of details for distant objects and less textured regions such as the wall. (Zoom in for the details.)

Table 1: Quantitative results in complex scenes.

| Methods | TAT | | | NeRF-360-v2 | | | Free-Dataset | | |
|---|---|---|---|---|---|---|---|---|---|
| | PSNR↑ | SSIM↑ | LPIPS↓ | PSNR↑ | SSIM↑ | LPIPS↓ | PSNR↑ | SSIM↑ | LPIPS↓ |
| NeRF++ | 20.419 | 0.663 | 0.451 | 27.211 | 0.728 | 0.344 | 24.592 | 0.648 | 0.467 |
| MipNeRF360 | **22.061** | **0.731** | **0.357** | **28.727** | **0.799** | **0.255** | **27.008** | **0.766** | 0.295 |
| MipNeRF360-short[*] | 20.078 | 0.617 | 0.508 | 25.484 | 0.631 | 0.452 | 24.711 | 0.648 | 0.466 |
| DVGO | 19.750 | 0.634 | 0.498 | 25.543 | 0.679 | 0.380 | 23.485 | 0.633 | 0.479 |
| Instant-NGP | 20.722 | 0.657 | 0.417 | 27.309 | 0.756 | 0.316 | 25.951 | 0.711 | 0.312 |
| F2-NeRF | – | – | – | 26.393 | 0.746 | 0.361 | 26.320 | **0.779** | **0.276** |
| Switch-NGP[†] | 20.512 | 0.654 | 0.432 | 26.524 | 0.740 | 0.331 | 25.755 | 0.694 | 0.341 |
| Block-NGP[†] | 20.783 | 0.659 | 0.415 | 27.436 | 0.761 | 0.298 | 26.015 | 0.702 | 0.325 |
| GML-NeRF | **21.708** | **0.672** | **0.398** | **27.871** | **0.769** | **0.298** | **26.449** | 0.719 | **0.285** |

[*] MipNeRF360 requires nearly one day for training. For a fair comparison, we also report its results with one hour training.

[†] We adapt Switch-NeRF and Block-NeRF to the Instant-NGP fast training framework.

recently SOTA single-NeRF framework-ZipNeRF (Barron et al., 2023), named GML-ZipNeRF, in Appendix G. As shown in Table 13 , GML-ZipNeRF obtains better rendering performance, validating GML-NeRF's potential for integration with different frameworks.

**GML-NeRF achieves better recovery of distant details and accurate rendering for less textured regions.** The qualitative results are shown in Figure 3. Compared to other methods, GML-NeRF achieves better rendering quality in both outdoor and indoor scenes. In outdoor scenes, GML-NeRF produces detailed and realistic rendering results for the sky, road and other distant objects. In indoor scenes, GML-NeRF generates more accurate details for less textured regions such as the wall. GML-NeRF takes advantage of the gate-guided parameters allocation in the ray dimension to effectively boost the model's performance. More results on the ScanNet dataset are shown in Appendix B.

**The gate module learns to reasonably assign gating scores.** We visualize how the gate module performs the allocation in the ray dimension in Figure 4. (1) In the Truck scene (left), the depth range between the center object and the background is large. The gate module assigns different preferences to sub-NeRF1 in these two regions and **distinguishes the foreground/background regions**. (2) In a more open scene with fewer occlusions (middle), the Playground scene, the gate module **allocates model parameters according to the direction of rays**. (3) In the Train scene

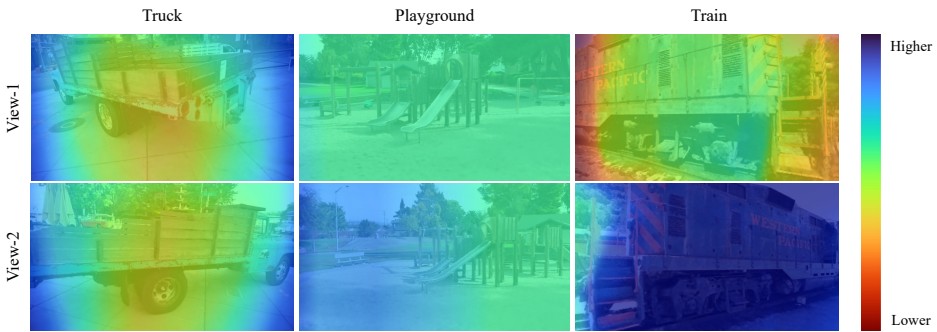

Figure 4: Visualization of the gating scores of sub-NeRF1 on two different views. GML-NeRF obtains different and reasonable parameter allocation in different types of scenes.

(right), sub-NeRF1 has different preferences for the two sides of the train. Such a scene is more like an object, where **assigning different model parameters to different sides of the object** helps improve the performance. This visualization explains why GML-NeRF also works well on simple object datasets. The above results demonstrate that GML-NeRF learns a reasonable parameter allocation across different types of scenes.

**Scaling up NeRF with the GML-NeRF framework is more effective than scaling the MLP width, increasing the feature grid size, or adding more feature grids.** By default, we set the number of sub-NeRFs to 2 in all experiments. As shown in Figure 5, when the number of sub-NeRFs increases, GML-NeRF consistently obtains average performance gains on the ScanNet dataset while only marginally increasing the number of model parameters. Compared with directly increasing the hidden dimension of MLP decoders or the size of feature grid, GML-NeRF has a better performance-model size scalability. See Appendix F for additional results on GML-NeRF's scalability.

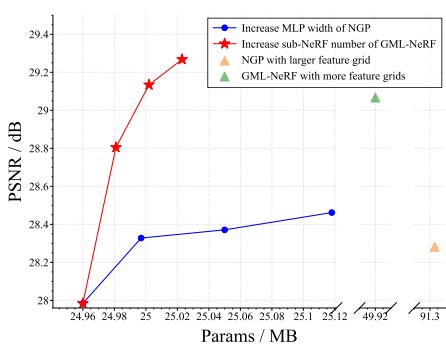

Figure 5: Scalability study of GML-NeRF.

### 5.3 Ablation Studies

In this section, we conduct ablation studies on GML-NeRF using the TAT dataset (Knapitsch et al., 2017). The key takeaways from our results are summarized bellow and some additional ablation studies and analysis are presented in the Appendix D.

**Importance of the gate-guided multi-NeRF fusion and depth-based mutual learning.** The ablation results of the two key components of GML-NeRF are shown in Table 2. Uniform fusion simply averages multi-NeRF outputs to get final rendering results without a gate module. In this way, all the sub-NeRFs focus on all the training rays instead of having their own preferences, which can not effectively improve rendering quality. For the depth-based mutual learning method, we observe that it helps provide a smoother and more reasonable depth prediction, as shown in Figure 6. In addition to improving rendering consistency, it also acts as a geometric regularization to reduce the ambiguity of geometry modeling and avoid overfitting.

**Importance of the ray-level allocation.** We evaluate the results of different fusion dimensions in Table 3. Compared to fusing multi-NeRFs' outputs in the point dimension, our ray-based method performs better, validating the superiority of the visibility-aware multi-NeRF method.

**Importance of pixel-granularity fusion.** We compare different fusion granularity in Table 4. In the image-granularity fusion, all pixels of an image have the same preference for model parameters, which may not be reasonable. An illustrative example is an image capturing both the central object and the background region, such as the *Truck* scene shown in Figure 4. In such a case, the rays hitting

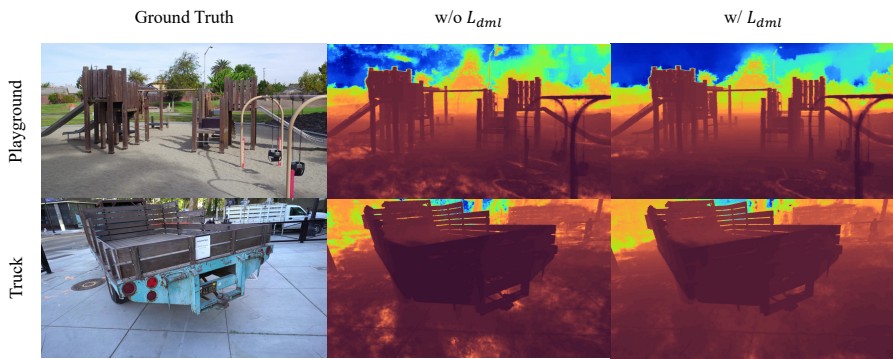

Figure 6: Depth visualization comparison between w/o $L_{dml}$ and w/ $L_{dml}$ on TAT dataset. Zoom in to see the details of sky and ground.

Table 2: Ablation results of gate-guided multi-NeRF fusion and depth-based mutual learning.

| Method | Metric | M60 | Playground | Train | Truck | Avg |
|---|---|---|---|---|---|---|
| Uniform fusion | PSNR↑ | 19.229 | 22.863 | 17.531 | 23.569 | 20.798 |
| | SSIM↑ | 0.633 | 0.694 | 0.596 | 0.746 | 0.667 |
| | LPIPS↓ | 0.431 | 0.414 | 0.451 | 0.345 | 0.411 |
| w/o depth mutual loss | PSNR↑ | 18.912 | 23.399 | 17.371 | 24.665 | 21.087 |
| | SSIM↑ | 0.621 | 0.694 | 0.589 | 0.758 | 0.666 |
| | LPIPS↓ | 0.436 | 0.402 | 0.449 | 0.329 | 0.404 |
| GML-NeRF | PSNR↑ | 19.051 | 23.901 | 19.369 | 24.509 | **21.708** |
| | SSIM↑ | 0.631 | 0.689 | 0.612 | 0.757 | **0.672** |
| | LPIPS↓ | 0.429 | 0.402 | 0.431 | 0.333 | **0.399** |

these two regions should be assigned different model parameters. In contrast, pixel-granularity fusion provides a more fine-grained understanding of the image and scene.

Table 3: Ablation results of fusion dimensions.

| Fusion Dimension | PSNR↑ | SSIM↑ | LPIPS↓ |
|---|---|---|---|
| Point-level | 20.796 | 0.661 | 0.413 |
| Ray-level (Ours) | **21.708** | **0.672** | **0.399** |

Table 4: Ablation results of fusion granularity.

| Fusion Granularity | PSNR↑ | SSIM↑ | LPIPS↓ |
|---|---|---|---|
| Image-level | 21.503 | 0.669 | 0.408 |
| Pixel-level (Ours) | **21.708** | **0.672** | **0.399** |

## 6 CONCLUSION AND OUTLOOKS

This work proposes a gate-guided mutual learning framework (GML-NeRF) for neural rendering. To alleviate the issue of limited model capacity in complex scenes, we construct a multi-NeRF framework and allocate model parameters in the ray dimension. This allocation is guided by a learnable soft gate module, allowing different sub-NeRFs to focus on specific rays rather than uniformly distributing their attention across all rays. Additionally, we propose a depth-based mutual learning method that improves the multi-NeRF rendering consistency and reduces the depth ambiguity, thereby improving generalization to novel views. Extensive experiments validate that GML-NeRF is effective and applicable to various types of datasets.

As GML-NeRF has showcased superior scaling performance than directly enlarging the MLP, we prospect its broader application in larger and more complex scenes in the future. Beyond scaling up NeRF for modeling more complex scenes, scaling up NeRF for enhanced generalization to novel scenes is of great application and research interests (Chen et al., 2021; Wang et al., 2021b). The adaption of GML-NeRF to cross-scene neural rendering is an interesting direction to explore for future work.

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

# A IMPLEMENTATION DETAILS

## A.1 IMPLEMENTATION DETAILS OF GML-NERF

**Architecture Details.** Our GML-NeRF is built upon Instant-NGP (Müller et al., 2022) using a third-party PyTorch implementation [1] and costs no more than one hour of training. We follow the original architecture of Instant-NGP with 16 levels of resolution. The hash table length at each resolution is fixed to $2^{19}$. The density and color MLP comprise one and two hidden layers with 64 channels respectively.

**Training Details.** For Instance-NGP and our GML-NeRF, we train the NeRFs for 20k iterations on a single RTX-3090 GPU. We use Adam optimizer with a batch size of 8192 rays and a learning rate decaying from $1 \times 10^{-2}$ to $3 \times 10^{-4}$. For the weights of the regularization terms in Equation 6, $\lambda_1$ is set to $1 \times 10^{-4}$ on NeRF-360-v2 and Free dataset, and is set to $5 \times 10^{-3}$ on other datasets. We set $\lambda_2$ to $1 \times 10^{-2}$ on all the datasets. By default, the number of sub-NeRFs is set to 2, and it is sufficient to achieve significant rendering quality improvement.

Some previous work has observed that the gate module tends to converge to an imbalanced state, where it always produces large weights for the same few sub-models (Shazeer et al., 2017b; Wang et al., 2022; Zhenxing & Xu, 2022). Such an imbalance problem exists in GML-NeRF as well. Once the gate module is trapped in a local optimum solution, it will always choose a specific sub-NeRF for rendering and can't effectively allocate model parameters in the ray dimension.

Following (Shazeer et al., 2017b; Wang et al., 2022), we adopt the regularization on the Coefficient of Variation of the soft gating scores, which encourages a balanced allocation of model parameters for training rays. The CV loss function is given by

$$L_{cv} = \frac{\text{Var}(\overline{G}(\mathcal{R}))}{\left(\sum_{k=1}^{n} \overline{G_k}(\mathcal{R})/n\right)^2},$$ 
(7)

$$\overline{G_k}(\mathcal{R}) = \sum_{\mathbf{r} \in \mathcal{R}} G_k(\mathbf{r}),$$ 
(8)

where $\overline{G}(\mathcal{R})$ is the set $\left\{\overline{G_k}(\mathcal{R})\right\}_{k=1}^{n}$. Note that some work also uses the load-balanced loss to encourage multi-models to receive roughly equal numbers of training examples (Shazeer et al., 2017b; Zhenxing & Xu, 2022). However, this optimization objective is too strict and unsuitable for our framework.

## A.2 IMPLEMENTATION DETAILS OF SWITCH-NGP

Switch-NeRF (Zhenxing & Xu, 2022) constructs a point-based multi-NeRF framework based on MLP-based NeRF structure. Given a 3D point $\boldsymbol{x}$, it first extracts high-level point feature $S(\boldsymbol{x})$ using a linear layer, which will be sent to the gate module to obtain the gating scores. Then, they apply a Top-1 function on the gating scores to determine which NeRF expert should be activated. The output feature of the chosen expert will be multiplied by the gating score corresponding to the expert and obtain the fused point feature. Finally, the fused point feature is sent to the unified MLPs to predict the density $\sigma$ and color $c$.

As illustrated in Figure 7, we build an NGP-version of Switch-NeRF, named Switch-NGP. Since NGP contains a feature grid in the form of the hash table, we directly use the feature grid to obtain the high-level point feature $S(\boldsymbol{x})$ of the point $\boldsymbol{x}$. Switch-NeRF has validated the importance of a unified head, wherein the gating score is multiplied by the high-level features rather than the density or color predictions, which makes the gating and prediction more stable in training. We also perform the multi-NeRF fusion in the point-feature dimension by inserting extra $K$ feature MLPs before the density MLP. Each expert in Switch-NGP corresponds to a tiny feature MLP with two hidden layers and 64 channels.

The training details of Switch-NGP is the same as GML-NeRF, as described in Section A.1.

---

[1] https://github.com/kwea123/ngp_pl

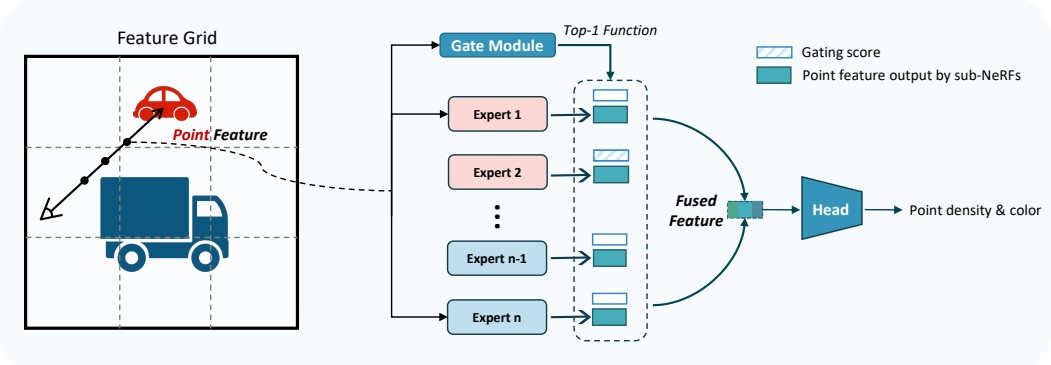

Figure 7: The overview of Switch-NGP.

### A.3 IMPLEMENTATION DETAILS OF BLOCK-NGP

Block-NeRF (Tancik et al., 2022) applies the multi-NeRF method to the street scene, which also allocates model parameters in the ray dimension but in the image-level granularity. Specifically, Block-NeRF places one NeRF at each intersection and directly allocates the training images to multi-NeRFs according to the image shooting positions. We implement an NGP-version Block-NeRF, named Block-NGP, which can be applied to various types of scenes without prior knowledge. After getting all the training images, we first use the clustering algorithm (KMeans) to cluster the image shooting positions, and the number of clusters is set the same as the number of sub-NeRFs. During the training process, each training image is allocated to the corresponding sub-NeRF according to the clustering results, and the training of sub-NeRFs is independent.

## B EXPERIMENTS ON SCANNET DATASET

We compare GML-NeRF with other multi-NeRF methods on the ScanNet dataset (Dai et al., 2017), an indoor scene RGB-D dataset. Compared to other outdoor datasets, ScanNet contains more texture-less regions like the floors and the walls, which poses more challenges for neural rendering. We conduct experiments in four complete scenes in ScanNet, namely scene0046, scene0276, scene0515 and scene0673. For each scene, we train NeRFs with the whole image set (more than one thousand images with 484 × 648 resolution for each scene) and test on one-sixteenth of the images. The quantitative and qualitative results are shown in Table 5 and Figure 8 respectively. Our GML-NeRF outperforms other multi-NeRF methods and renders less blur.

Table 5: Quantitative results on ScanNet dataset.

| Methods | Metrics | scene0046 | scene0276 | scene0515 | scene0673 | Avg |
|---|---|---|---|---|---|---|
| NGP | PSNR↑ | 28.504 | 29.996 | 28.159 | 25.278 | 27.984 |
| | SSIM↑ | 0.839 | 0.835 | 0.786 | 0.686 | 0.786 |
| | LPIPS↓ | 0.413 | 0.421 | 0.448 | 0.472 | 0.438 |
| Switch-NGP | PSNR↑ | 28.135 | 29.614 | 27.814 | 25.140 | 27.676 |
| | SSIM↑ | 0.834 | 0.831 | 0.779 | 0.684 | 0.782 |
| | LPIPS↓ | 0.421 | 0.431 | 0.456 | 0.473 | 0.445 |
| Block-NGP | PSNR↑ | 28.728 | 30.214 | 28.332 | 25.444 | 28.180 |
| | SSIM↑ | 0.842 | 0.840 | 0.789 | 0.688 | 0.790 |
| | LPIPS↓ | 0.408 | 0.416 | 0.443 | 0.469 | 0.434 |
| GML-NeRF | PSNR↑ | 29.440 | 30.871 | 29.149 | 25.759 | **28.805** |
| | SSIM↑ | 0.851 | 0.843 | 0.800 | 0.690 | **0.796** |
| | LPIPS↓ | 0.396 | 0.405 | 0.427 | 0.469 | **0.424** |

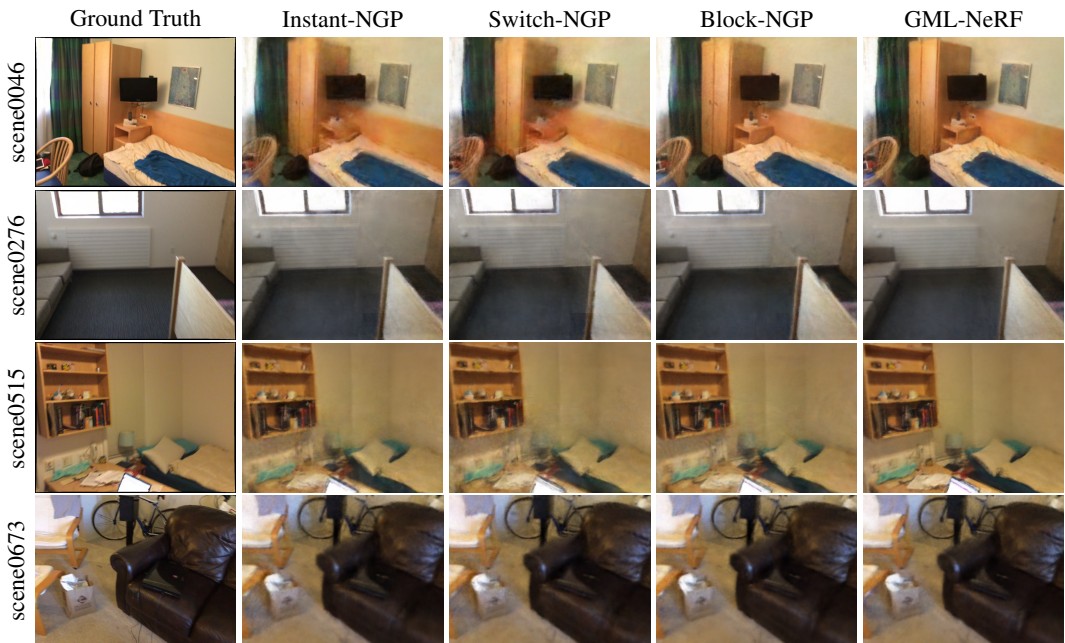

Figure 8: Qualitative comparisons on ScanNet dataset. Compared to other multi-NeRF methods, GML-NeRF renders less blur and achieves better recovery of details.

## C    PER-SCENE RESULTS

We provide the per-scene quantitative results on the Mask-TAT dataset, TAT dataset, NeRF-360-v2 dataset and Free dataset in Table 6, Table 7, Table and Table respectively. The results are reported in the metric of PSNR.

Table 6: Scene breakdown on the Mask-TAT dataset.

| Methods | Ignatius | Truck | Barn | Caterpillar | Family | Avg |
|---|---|---|---|---|---|---|
| NeRF | 25.43 | 25.36 | 24.05 | 23.75 | 30.29 | 25.78 |
| MipNeRF | 29.037 | 23.19 | 28.481 | 28.016 | 29.009 | 27.547 |
| PlenOctrees | 28.19 | 26.83 | 26.8 | 25.29 | 32.85 | 27.99 |
| DVGO | 28.16 | 27.15 | 27.01 | 26.00 | 33.75 | 28.41 |
| Instant-NGP | 28.431 | 27.562 | 27.611 | 26.065 | 34.092 | 28.752 |
| Switch-NGP | 28.184 | 27.34 | 27.472 | 25.75 | 33.711 | 28.491 |
| Block-NGP | 28.202 | 27.621 | 27.768 | 26.06 | 34.081 | 28.746 |
| GML-NeRF | 29.806 | 28.163 | 28.701 | 27.445 | 34.756 | 29.774 |

Table 7: Scene breakdown on the TAT dataset.

| Methods | M60 | Playground | Train | Truck | Avg |
|---|---|---|---|---|---|
| NeRF | 16.86 | 21.55 | 16.64 | 20.85 | 18.975 |
| NeRF++ | 17.964 | 22.914 | 18.194 | 22.603 | 20.419 |
| MipNeRF-360 | 20.091 | 24.27 | 19.741 | 24.144 | 22.062 |
| MipNeRF360short | 18.394 | 22.682 | 17.738 | 21.497 | 20.078 |
| DVGO | 17.292 | 22.62 | 17.783 | 21.306 | 19.750 |
| Instant-NGP | 18.914 | 22.832 | 17.707 | 23.428 | 20.720 |
| Switch-NGP | 18.619 | 22.661 | 17.523 | 23.243 | 20.512 |
| Block-NGP | 18.879 | 22.555 | 18.048 | 23.651 | 20.783 |
| GML-NeRF | 19.051 | 23.901 | 19.369 | 24.509 | 21.708 |

Table 8: Scene breakdown on the NeRF-360-v2 dataset.

| Methods | bicycle | bonsai | counter | garden | kitchen | room | stump | Avg |
|---|---|---|---|---|---|---|---|---|
| NeRF | 21.818 | 29.028 | 26.980 | 23.640 | 27.164 | 30.097 | 22.934 | 25.952 |
| NeRF++ | 21.426 | 31.67 | 27.717 | 24.801 | 29.468 | 30.621 | 24.770 | 27.210 |
| MipNeRF360 | 22.861 | 32.97 | 29.291 | 26.014 | 31.987 | 32.685 | 25.278 | 28.727 |
| MipNeRF360short | 21.264 | 28.040 | 26.366 | 23.214 | 26.552 | 29.636 | 23.313 | 25.484 |
| DVGO | 21.652 | 27.919 | 26.432 | 23.851 | 26.282 | 31.677 | 20.988 | 25.543 |
| F2-NeRF | 21.311 | 30.036 | 25.873 | 23.694 | 28.935 | 29.421 | 24.251 | 26.217 |
| Instant-NGP | 24.203 | 31.374 | 25.665 | 25.312 | 30.278 | 31.534 | 22.799 | 27.309 |
| Switch-NGP | 23.859 | 30.012 | 24.359 | 25.164 | 29.865 | 31.127 | 21.284 | 26.524 |
| Block-NGP | 24.186 | 31.684 | 25.704 | 25.288 | 30.382 | 31.569 | 23.241 | 27.436 |
| GML-NeRF | 24.550 | 32.439 | 25.230 | 25.634 | 31.062 | 32.863 | 23.312 | 27.871 |

Table 9: Scene breakdown on the Free dataset

| Methods | Hydrant | Lab | Pillar | Road | Sky | Stair | Grass | Avg |
|---|---|---|---|---|---|---|---|---|
| NeRF | 16.569 | 17.342 | 20.944 | 19.793 | 15.925 | 18.731 | 22.439 | 18.820 |
| NeRF++ | 22.948 | 23.718 | 26.353 | 24.916 | 25.059 | 27.647 | 21.504 | 24.592 |
| MipNeRF360 | 25.03 | 26.57 | 29.22 | 27.07 | 26.99 | 29.79 | 24.39 | 27.008 |
| MipNeRF360short | 23.281 | 24.412 | 26.789 | 24.158 | 25.369 | 27.139 | 21.827 | 24.711 |
| DVGO | 22.315 | 23.123 | 25.345 | 23.242 | 24.736 | 25.844 | 19.794 | 23.485 |
| Instant-NGP | 23.29 | 26.084 | 28.683 | 26.302 | 26.05 | 28.158 | 23.088 | 25.951 |
| F2-NeRF | 24.34 | 25.92 | 28.76 | 26.76 | 26.41 | 29.19 | 22.87 | 26.32 |
| Switch-NGP | 23.197 | 25.901 | 28.08 | 26.155 | 26.034 | 28.097 | 22.819 | 25.755 |
| Block-NGP | 23.663 | 26.682 | 28.103 | 25.989 | 26.283 | 28.395 | 22.988 | 26.015 |
| GML-NeRF | 24.463 | 25.751 | 28.871 | 26.827 | 27.235 | 28.562 | 23.433 | 26.449 |

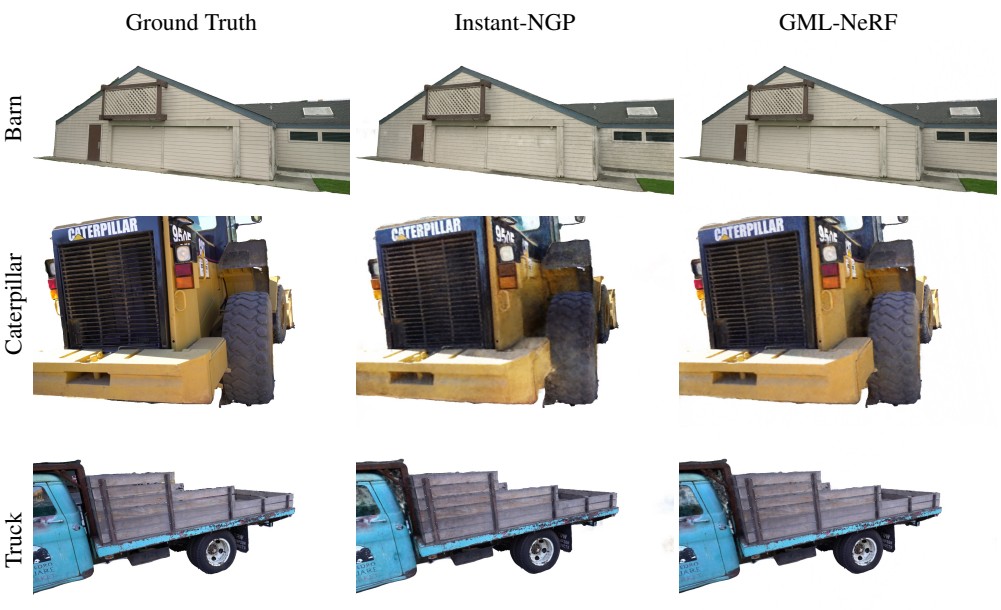

Figure 9: Qualitative comparisons on the MaskTAT dataset.

Table 10: Additional ablation results.

| Method | Metric | M60 | Playground | Train | Truck | Avg |
|---|---|---|---|---|---|---|
| Equal DML | PSNR↑ | 18.929 | 23.108 | 19.012 | 24.625 | 21.419 |
| | SSIM↑ | 0.625 | 0.686 | 0.610 | 0.758 | 0.670 |
| | LPIPS↓ | 0.431 | 0.405 | 0.432 | 0.332 | 0.400 |
| Independent feature grids | PSNR↑ | 18.765 | 22.839 | 18.958 | 24.493 | 21.264 |
| | SSIM↑ | 0.625 | 0.697 | 0.614 | 0.762 | 0.675 |
| | LPIPS↓ | 0.431 | 0.405 | 0.417 | 0.325 | 0.395 |
| Uniform fusion w/o DML | PSNR↑ | 19.229 | 22.863 | 17.531 | 23.569 | 20.798 |
| | SSIM↑ | 0.633 | 0.694 | 0.596 | 0.746 | 0.667 |
| | LPIPS↓ | 0.431 | 0.414 | 0.451 | 0.345 | 0.411 |
| Uniform fusion w/ DML | PSNR↑ | 19.005 | 22.766 | 17.532 | 23.513 | 20.704 |
| | SSIM↑ | 0.627 | 0.695 | 0.592 | 0.747 | 0.665 |
| | LPIPS↓ | 0.434 | 0.411 | 0.453 | 0.341 | 0.410 |
| w/o CV loss | PSNR↑ | 18.743 | 22.795 | 17.245 | 23.395 | 20.545 |
| | SSIM↑ | 0.619 | 0.683 | 0.587 | 0.731 | 0.655 |
| | LPIPS↓ | 0.445 | 0.419 | 0.465 | 0.354 | 0.421 |
| Half CV loss | PSNR↑ | 19.114 | 24.003 | 19.462 | 24.518 | 21.774 |
| | SSIM↑ | 0.625 | 0.689 | 0.606 | 0.758 | 0.670 |
| | LPIPS↓ | 0.433 | 0.404 | 0.430 | 0.334 | 0.400 |
| GML-NeRF | PSNR↑ | 19.051 | 23.901 | 19.369 | 24.509 | 21.708 |
| | SSIM↑ | 0.631 | 0.689 | 0.612 | 0.757 | 0.672 |
| | LPIPS↓ | 0.429 | 0.402 | 0.431 | 0.333 | 0.399 |

## D ADDITIONAL ABLATION STUDIES

We add additional ablation studies on the TAT dataset to further analyze the mechanism of GML-NeRF, including structural design, depth-mutual learning and CV balanced regularization. The results are shown in Table 10.

**Gate-guided depth mutual learning.** In GML-NeRF, we use the gate-guided fused depth as the target depth to regularize sub-NeRFs' geometry and avoid overfitting. By contrast, when we directly use the average of the sub-NeRFs' rendering depths as the target depth, which means all sub-NeRFs have equal regularization strength (Equal DML), the rendering quality will be slightly worse. The results highlights the pivotal role of gate-guided depth mutual learning. Using the gated-guided fused depth as the target depth differently penalizes sub-depths based on the gating scores and increase the accuracy of the geometry regularization. We also observe that depth mutual learning has no effect in the case of uniform fusion due to the low accuracy of the averaged depth.

**Structural design.** In GML-NeRF, we adopt a multi-NeRF structure with a shared feature grid and an ensemble of MLP-decoders. We further analyze the reason behind the performance improvement and explore the performance of independent feature grids. As Table 10 shows, the model employing a shared feature grid (GML-NeRF) outperforms its counterpart with multiple independent feature grids, which highlights the effect of independent MLP decoders rather than feature grid. We attribute this observation and the performance gained by GML-NeRF into two aspects. (1)Within the hybrid representation, the feature grid is responsible for encoding features of 3D spatial points, while the MLP encoder is designed to encode view information. The crucial design of independent MLP decoders aligns with our visibility-aware motivation, thereby enhancing the view-dependent effect. (2)The training complexity will also increase as the trainable parameters increase. With the limited amount of training data, increasing the number of feature grids leads to poor convergence. By contrast, as different rays may pass through the same region of 3D space, weight sharing for the feature grid helps to facilitate training. Although the number of learnable parameters hardly increases, GML-NeRF achieves a more optimal capacity allocation in the ray dimension, helping to increase the model's generalization ability.

**CV balanced regularization.** As introduced in Section A.1, we adopt the regularization on the Coefficient of Variation of the soft gating scores to prevent the gate module from collapsing onto

a specific sub-NeRF while maintaining sub-NeRF's different specialties. Without CV balanced regularization, the rendering quality degrades significantly. Besides, we apply the CV regularization only for the first half of the training time and find that the performance is comparable to GML-NeRF, The results prove that such regularization would not interfere with the learning of the gate module.

# E    DISCUSSION OF MEGA-NGP

Mega-NeRF (Turki et al., 2022) applies the multi-NeRF method to the drone scenes, allocating model parameters in the ray dimension and the pixel-level granularity. Specifically, it allocates rays by partitioning the intersecting points between rays and scenes. Such a method is suitable for drone scenes, where the top-down perspective allows for the approximation of ray-scene intersections by intersecting with a set horizontal plane. However, in unstructured scenes captured by free trajectories, the intersecting points between rays and scenes cannot be determined before the training is completed, limiting the applicability of Mega-NeRF to such scenes.

Since there is no straightforward implementation to determine the ray intersections before training, we adopt an alternative implementation for NGP-version Mega-NeRF, which employs a clustering algorithm to divide rays directly based on their origins and directions. The clustering process is offline and same as the one in Block-NGP. During the training process, each training pixel is allocated to one corresponding sub-NeRF according to the clustering results. To ensure a fair comparison, the model structure of Mega-NGP is the same as the one in GML-NeRF, following the implementation of Block-NGP. We conduct a comprehensive evaluation across all datasets and the experimental results are shown in Table 11. Mega-NGP yields similar results to Block-NGP, which is less effective than our GML-NeRF.

Table 11: Comparison with Mega-NGP and GML-NeRF

| Method | Metric | TAT | 360v2 | Free Dataset | ScanNet |
|---|---|---|---|---|---|
| Mega-NGP | PSNR↑ | 20.843 | 27.482 | 25.855 | 28.1 |
| | SSIM↑ | 0.659 | 0.761 | 0.696 | 0.786 |
| | LPIPS↓ | 0.415 | 0.311 | 0.332 | 0.437 |
| GML-NeRF | PSNR↑ | 21.708 | 27.87 | 26.449 | 28.87 |
| | SSIM↑ | 0.672 | 0.769 | 0.719 | 0.797 |
| | LPIPS↓ | 0.399 | 0.298 | 0.285 | 0.424 |

# F    MORE SCALABILITY STUDIES

We provide the per-scene results of scalability studies on the ScanNet dataset in Table 12 which are reported in the metric of PSNR.

Table 12: Scene breakdown of scalability studies on the ScanNet dataset.

| Method | 004600 | 027600 | 051500 | 067304 | Avg |
|---|---|---|---|---|---|
| Instant-NGP | 28.504 | 29.996 | 28.159 | 25.278 | 27.984 |
| GML-NeRF-size2 | 29.440 | 30.871 | 29.149 | 25.759 | 28.805 |
| GML-NeRF-size3 | 29.878 | 31.242 | 29.470 | 25.944 | 29.134 |
| GML-NeRF-size4 | 30.018 | 31.31 | 29.679 | 26.063 | 29.268 |

Furthermore, we observe that the model with four sub-NeRFs converges faster than the one with two sub-NeRFs while achieving better rendering quality with the same training iterations, as Figure 10 shows. The ease of training convergence can be attributed to two aspects. On the one hand, the feature grid is shared among multi-NeRFs, and thus, the number of learnable parameters increases marginally. On the other hand, as the neural network is better at fitting low-frequency information,

our gate module (a 4-layer MLP without sinusoidal position encoding) has implicitly incorporated "smoothness prior", leading to closer rays to be more possibly assigned closer gating scores.

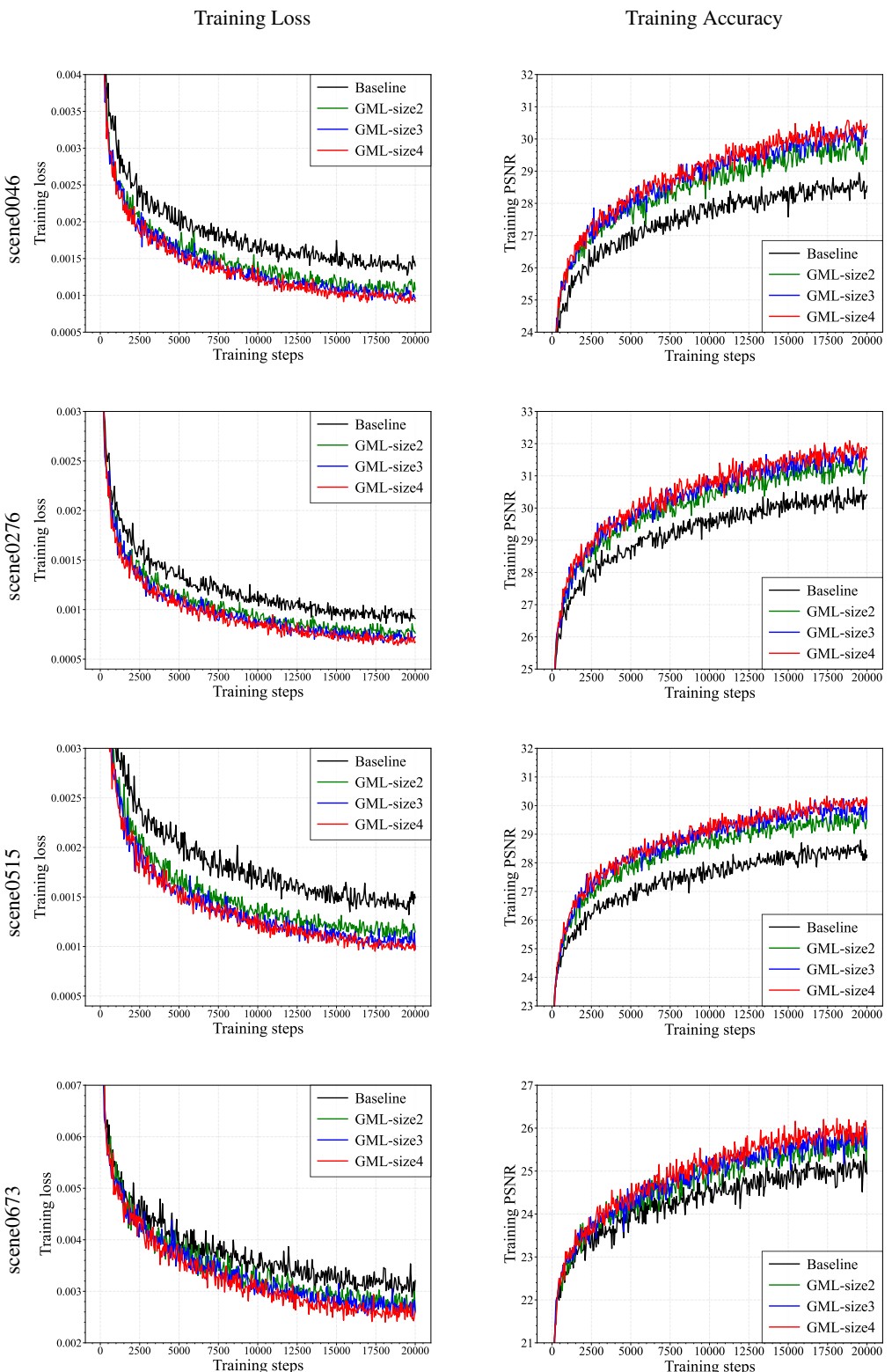

Figure 10: Convergence curve on the ScanNet dataset.

## G    INTEGRATION OF GML-NERF AND ZIP-NERF

As a multi-NeRF training framework, GML-NeRF is essentially orthogonal to the structure and training method of single-NeRF. For the benefit of training efficiency and its wide application, we build and validate GML-NeRF upon the Instant-NGP. Nevertheless, it can also be integrated with other single-NeRF frameworks, such as ZipNeRF (Barron et al., 2023) (a SOTA single-NeRF framework).

We implement a ZipNeRF version of GML-NeRF, named GML-ZipNeRF, and evaluate the performance on the 360v2 dataset. Similar to GML-NeRF, GML-ZipNeRF adopts a shared feature grid and multiple MLP decoders. The training settings are kept the same as the original paper, including the training iterations and batch size. As shown in Table 13, integrated with GML-NeRF, ZipNeRF can also obtain performance gains, validating GML-NeRF's effectiveness and potential for integration with different frameworks.

Considering that different frameworks have different characteristics, researchers may choose different frameworks based on specific situational requirements. Adapting GML-NeRF to different single-NeRF frameworks remains an interesting point to be explored in the future.

Table 13: Comparison with ZipNeRF and GML-ZipNeRF.

| Methods | bicycle | bonsai | counter | garden | kitchen | room | stump | Avg |
|---|---|---|---|---|---|---|---|---|
| ZipNeRF | 21.019 | 33.052 | 25.982 | 24.330 | 32.843 | 34.777 | 25.406 | 28.201 |
| GML-ZipNeRF | 20.488 | 33.486 | 26.372 | 24.603 | 33.120 | 35.795 | 25.581 | 28.492 |

