# OpenReview forum: "GML-NeRF: Gate-guided Mutual Learning Framework for Neural Rendering"
_ICLR.cc/2024/Conference — Submitted to ICLR 2024_

### Official Review · Reviewer_Vfdc · 2023-10-31

**Soundness:** 2 fair
**Presentation:** 3 good
**Contribution:** 2 fair
**Rating:** 5
**Confidence:** 4

**Summary:**

The paper introduces GML-NeRF, a new multi-NeRF approach designed to enhance the model's capacity for effectively representing complex scenes. GML-NeRF utilizes a hybrid NeRF architecture that combines a shared 3D dense grid parametrization, known as Instant-NGP, with multiple independent MLP decoders.

The proposed approach enables the prediction of color and density values associated with each camera ray by employing a weighted sum of independent MLP predictions, essentially creating a 'mixture of experts.' The weights for this summation are determined by a separate MLP with a Softmax output layer, called soft gating module. The author's findings indicate that GML-NeRF outperforms other multi-NeRF methods, delivering higher rendering quality for large-scale scenes.

**Strengths:**

The ray-based multi-NeRF approach distributes training rays across various NeRF models, offering greater flexibility compared to point-based multi-NeRF techniques. The GML-NeRF approach proposed in this study demonstrates enhanced effectiveness through the increase in the number of MLPs rather than merely widening the capacity of a single MLP. This method is particularly versatile and applicable to a wide range of scenes, regardless of whether they lack prior, scene-specific knowledge.

**Weaknesses:**

The paper exhibits several notable weaknesses. First, the soft gate module's mechanism is not comprehensively explored. It remains unclear whether the estimated scores are primarily based on scene geometry or if they serve mainly for the minimization of balancing the regularization term within the loss function.

Additionally, there appears to be a discrepancy between the motivating concept presented in Figure 1 (b) and the ray allocations depicted in Figure 4 (left), where rays are equally distributed between two NeRFs (i.e., MLPs). This contradiction raises questions about the method's consistency and its alignment with the initially proposed approach.

Lastly, the results obtained do not demonstrate a significant improvement when compared to other existing methods. This raises concerns about the overall effectiveness of the proposed technique.

**Questions:**

The balancing regularization aims to enforce equal weighting between MLPs. However, it seems that the assignment should ideally be based on scene geometry. This raises a concern, as your visualization in Figure 4 (left) appears counterintuitive, showing equal weights between two MLPs for the body of the truck. Can you provide clarification on this matter?

Regarding the ablation study on uniform fusion, it's not entirely clear whether you trained the model with the gating module and then computed the average between MLPs during testing, or if you trained the model without the gating module from the beginning. Can you please clarify on this process for a better understanding of your methodology?

---

> ### Author Response · Authors · 2023-11-19
> **Response to Reviewer Vfdc**
>
> We thank the reviewer for the constructive feedback and thoughtful comments. We address the detailed questions and comments below.
>
> **R4-Q1: Demonstration of Figure 4 and CV balance regularization**
>
> Thank the reviewer for the questions. We give a more detailed description of Figure 4 and CV balanced regularization below to clarify how GML-NeRF conducts the CV balanced regularization.
>
> (1) In Figure 4, the visualizations of gating scores in three scenes depict **the specialties of one sub-NeRF (sub-NeRF 1) for two different regions** (e.g., the body of the truck and other regions), instead of the weights of two sub-NeRFs.
>
> (2) We clarify that the CV balanced regularization is not to enforce equal weighting between sub-NeRFs. The CV loss is calculated on the average gating score within a batch rather than being applied to each ray individually. In this way, **it helps to prevent the gate module from collapsing onto a specific sub-NeRF on all rays while maintaining sub-NeRFs' specialties**. We highlight this mechanism in Section 4.4 in the revision.
>
> **R4-Q2: Discussion of soft gate module**
>
> Thank the reviewer for the insightful question. As clarified above, the CV balanced regularization is to prevent the gate module from collapsing onto a specific sub-NeRF and would not interfere with the learning of the gate module. The visual representation in Figure 4 reinforces this point by illustrating how the gate module learns a reasonable parameter allocation across different scenes. To further support this point, we conduct an ablation study of CV balanced regularization on the TAT dataset and add the results to the Appendix D. First of all, the rendering quality degrades without CV balanced loss, proving its necessity. Then, we apply the CV regularization only for the first half of the training time and find that the performance is comparable to that of GML-NeRF.
>
> | Method     | Metric | M60    | Playground | Train  | Truck  | Avg    |
> |------------|:------:|:------:|:----------:|:------:|:------:|:------:|
> | w/o CV     | PSNR   | 18.743 | 22.795     | 17.245 | 23.395 | 20.545 |
> | Half CV    | PSNR   | 19.114 | 24.003     | 19.462 | 24.518 | 21.774 |
> | Ours-size2 | PSNR   | 19.051 | 23.901     | 19.369 | 24.509 | 21.708 |
>
> **R4-Q3: Performance of GML-NeRF**
>
> Thank the reviewer for the consideration of performance. Overall, compared to Instant-NGP baseline, our scheme has improvements in the PSNR metric across various datasets: 1.02 on mask-TAT, 0.98 on TAT, 0.57 on 360v2, 0.49 on free dataset and 0.82 on ScanNet dataset. All the improvements are achieved by the default configuration of two sub-NeRFs. Notably, as the number of sub-NeRFs increases, an additional 0.4 PSNR gain is observed on the ScanNet dataset. We also check the performance of recently published and related work, such as F2-NeRF, which also implements a multi-NeRF framework on Instant-NGP. F2-NeRF claims the PSNR metric improvements of 0.15 on 360v2 and 1.91 on free dataset, which is on par with GML-NeRF effect. Additionally, on the 360v2 dataset, ZipNeRF obtains 0.74 PSNR gain compared to MipNeRF360, and 3D Gaussian Splatting-7k obtains 0.3 PSNR gain compared to INGP.
>
> Furthermore, GML-NeRF goes beyond comparison with existing SOTA single-NeRF methods, extending to the comparison with other multi-NeRF training frameworks based on NGP. In these comprehensive evaluations, GML-NeRF shows superiority over other multi-NeRF training methods.
>
> Finally, GML-NeRF acts as a multi-NeRF training method, which is adaptable to various types of scenes and compatible with different single-NeRF frameworks. Extensive experiments across complex scenes presented in the main paper affirm its adaptable ability.  Additionally, we implement the GML-ZipNeRF framework, which is a combined version of GML-NeRF and ZipNeRF (SOTA single-NeRF method). The experimental results prove GML-NeRF's effectiveness and potential for integration with different frameworks, as detailed in the response to reviewer PcY6 (R2-Q1) and included in the Appendix.
>
> **R4-Q4: Details of uniform fusion in Table 2**
> For the uniform fusion, we remove the soft gate module and just average sub-NeRFs' outputs in both training and inference time. Without the gating mechanism introduced by the gate module, sub-NeRFs lose the ray specialties, resulting in performance degradation.

---

> ### Author Response · Authors · 2023-11-21
> **Follow-up**
>
> Dear Reviewer Vfdc,
>
> Thanks once again for your valuable time in reviewing our paper. Did our previous responses address your concerns satisfactorily? If you have any further questions, please kindly inform us. We are more than willing to provide further clarifications.
>
> Authors

---

> ### Author Response · Authors · 2023-11-23
> **Looking forward to the reply**
>
> Dear reviewer Vfdc,
>
> Thank you once more for the review. We eagerly anticipate your feedback on our response. Did our previous response and revision address all your concerns? If you have any further questions, please let us know and we will try our best to provide additional clarification before the DDL.
>
> Authors

---

### Official Review · Reviewer_3Q2w · 2023-11-01

**Soundness:** 2 fair
**Presentation:** 2 fair
**Contribution:** 3 good
**Rating:** 5
**Confidence:** 2

**Summary:**

This paper proposes to ensemble sub-NeRFs on complex scenes by a trainable ray-based gate. The gate module learns the preference of sub-NeRF for each ray. The results of different sub-NeRFs are fused with the gating score. A depth-based mutual learning method is proposed to enhance the rendering consistency among multiple sub-NeRFs.

**Strengths:**

1. The motivation of learning ray-based allocation for different sub-NeRFs is interesting.
2. It designs a multiple NeRF network based on Instant-NGP to validate the learning-based ray allocation.
3. A depth mutual learning loss is proposed to align the depth of different sub-NeRFs to avoid overfitting.
4. It adapts several multi-NeRF methods to validate its core designs.

**Weaknesses:**

I have several concerns about the network design and results.

1. One of the motivations of this paper is that complex scenes need more network parameters and the parameters need to be allocated in the ray dimension. In the network design, the hash table is shared. The sub-NeRFs only contain 2 small decoders. Since in Instant-NGP, the decoder only contains very few parameters than the hash table (no more than 1%), it is unclear to me why the proposed method gets very good quality improvement with a minor increase of parameters.
2. The Block-NeRF uses a fixed and image-level ray allocation. As stated in paper A.3, the training of sub-NeRFs in Block-NGP is independent. So I assume the hash tables are not shared in Block-NGP. Therefore, Block-NGP will have 2 large hash tables. It is unclear to me why it only gets a very small improvement than Instant-NGP (20.783 vs 20.722 on TAT, 26.015 vs 25.951). Therefore, I think the detailed training of Block-NGP should be provided. In the paper of Block-NeRF, it also exploits the interpolation of results of sub-NeRFs in the inference. This paper should also provide details of the inference of Block-NGP.
3. For Table 4 of Importance of pixel-granularity fusion, it is unclear how the Image-level fusion is performed.
4. For Table 3 of Importance of the ray-level allocation, when using "fusing multi-NeRFs’ outputs in the point dimension", it is unclear how the point outputs are fused. What kind of gate is used for point fusion? Is it a point-based or ray-based gate?
5. Since this method learns pixel-level ray allocation, this method needs to directly compare the fixed pixel-level ray allocation methods. The Mega-NeRF actually allocates rays across different sub-NeRFs although the ray-allocation is determined by 3D points. The sub-NeRFs in the Mega-NeRF are trained independently and the results are interpolated by 3D distance. The hash tables may not be shared in Mega-NeRF. However, it can still be compared by setting the proposed method as not sharing hash tables. Since the image-level Block-NeRF is compared, I think the pixel-level Mega-NeRF is also worth comparison.
6. In Table 2, what is the accuracy if we train Uniform fusion with depth mutual loss?
7. The results of other datasets without depth mutual loss are needed to better analyze the effects of gate and depth mutual loss.
8. What is the result if Switch-NGP uses all the sub-NeRF outputs weighted by gating scores? This can actually be seen as a learnable point-based allocation method aligning with the training setting of the proposed method.

**Questions:**

My questions are in the Weaknesses.

---

> ### Author Response · Authors · 2023-11-19
> **Response to Reviewer 3Q2w-1**
>
> We thank the reviewer for the constructive feedback and thoughtful comments. We address the detailed questions and comments below.
>
> **R3-Q1: Analysis of the performance improvement**
>
> Thank the reviewer for the insightful question. We conduct an ablation study in the response to reviewer MyQ8 (R1-Q3), and **the results highlight the effect of independent MLP decoders rather than feature grid**. Within the hybrid representation, the explicit part (feature grid) encodes the features of 3D spatial points, while the implicit representation encodes the ray information. As our motivation is to consider the different visibility of a target region to different views, using multiple MLP encoders with different ray specialties is reasonable and effective. Although the number of learnable parameters hardly increases, GML-NeRF achieves a more optimal capacity allocation in the ray dimension, which helps to increase the model's generalization ability. Additionally, as different rays may pass through the same region of 3D space, weight sharing for the feature grid helps to facilitate training.
>
> **R3-Q2: Details and discussion of Block-NGP**
>
> For a fair comparison, the structure of the model in Block-NGP is the same as the one employed in GML-NeRF.  Despite implementing the same model structure, Block-NGP does not obtain performance improvements compared to the baseline (Instant-NGP). It's essential to clarify that this lack of improvement is not attributed to the model's structural design. As mentioned in response to R3-Q1, the pivotal design in GML-NeRF lies in the use of multiple MLP encoders with different ray specialties rather than the feature grid. Such assertion is also supported by the ablation studies addressed in response to reviewer MyQ8 (R1-Q3).
>
> The reason behind Block-NGP's poor results primarily stems from three aspects. Firstly, the manual image allocation scheme based on image position is not suitable for the scenes captured with arbitrary trajectories. Despite implementing the allocation through the K-means cluster algorithm, the allocation result may not be reasonable. Secondly, compared to the discrete 3D points, different rays have complex spatial relationships, such as crossing or passing through the same region, which makes Block-NGP's "hard" allocation ineffective. Thirdly, Block-NGP implements a view-level gating module, outputting a score to each image, which is a coarser granularity compared to pixel-level fusion.
>
> Furthermore, in the inference time, Block-NGP doesn't perform the interpolation of results from sub-NeRFs. This implementation is due to the inherent difficulty in defining a proper interpolation method when dealing with various complex scenes captured by free trajectories.
>
> **R3-Q3: Implementations of ablation studies**
>
> Thank the reviewer for pointing out the confusion in Tables 3 & 4. We appreciate the opportunity to provide clarification on the confusion.
>
> (1) Ablation Study of Fusion Dimension (Table 3): In this ablation study, we explore the superiority of GMl-NeRF over the point-based multi-NeRF method. The point-based method adopts a model structure similar to GML-NeRF, incorporating a shared feature grid and a soft gate module. The difference is that the soft gate module encodes the 3D points' data and outputs a gating score for each point. The fusion process is completed on the point dimension, followed by volume rendering.
>
> (2) Ablation Study of Fusion Granularity (Table 4): In this ablation study, both experiments are conducted within the GML-NeRF framework. For the image-level fusion, we directly average the ray directions of all pixels in each image to obtain the image directions. Then, the gate module outputs a score to each image, which is a coarser granularity compared to pixel-level fusion.

---

> > ### Author Response · Authors · 2023-11-19
> > **Response to Reviewer 3Q2w-2**
> >
> > **R3-Q4: Discussion of Mega-NGP**
> >
> > Mega-NeRF allocates rays by partitioning the intersecting points between rays and scenes. Such a method is suitable for drone scenes, where the top-down perspective allows for the approximation of ray-scene intersections by intersecting with a set horizontal plane.
> >
> > However, in unstructured scenes captured by free trajectories, the intersecting points between rays and scenes cannot be determined before the training is completed, limiting the applicability of Mega-NeRF to such scenes. In contrast, GML-NeRF's learnable allocation makes it versatile and applicable to various scenes.
> >
> > Since there is no reasonable implementation to determine the ray intersections before training, we adopt an alternative implementation, which employs a clustering algorithm to divide rays directly based on their origins and directions. The implementation details are demonstrated in the Appendix E. To ensure a fair comparison, the model structure of Mega-NGP is the same as the one in GML-NeRF, following the implementation of Block-NGP. We conduct a comprehensive evaluation across all datasets and add the experimental results in the Appendix E. Mega-NGP yields similar results to Block-NGP, which is less effective than our GML-NeRF.
> >
> > | Method   | Metric | TAT    | 360v2  | Free Dataset | ScanNet |
> > |----------|:------:|:------:|:------:|:------------:|:-------:|
> > | Mega-NGP | PSNR   | 20.843 | 27.482 | 25.855       | 28.100  |
> > |          | SSIM   | 0.659  | 0.761  | 0.696        | 0.786   |
> > |          | LPIPS  | 0.415  | 0.311  | 0.332        | 0.437   |
> > | GML-NeRF | PSNR   | 21.708 | 27.870 | 26.449       | **28.805**  |
> > |          | SSIM   | 0.672  | 0.769  | 0.719        | **0.796**   |
> > |          | LPIPS  | 0.399  | 0.298  | 0.285        | **0.424**   |
> >
> > **R3-Q5: Effect of uniform fusion + depth mutual loss**
> >
> > Thank the reviewer for the advice. We add the results of training with uniform fusion and depth mutual loss in the Appendix D. Without gate-guided fusion, sub-NeRFs lose their ray specialties, resulting in an ineffective improvement in rendering quality. Additionally, we also find that depth mutual learning is not helpful in this case, possibly due to the low accuracy of the fused depth.
> >
> > | Method                 | Metric | M60    | Playground | Train  | Truck  | Avg    |
> > |------------------------|:------:|:------:|:----------:|:------:|:------:|:------:|
> > | uniform fusion w/o dml | PSNR   | 19.229 | 22.863     | 17.531 | 23.569 | 20.798 |
> > | uniform fusion w/ dml  | PSNR   | 19.005 | 22.766     | 17.532 | 23.513 | 20.704 |
> > | Ours.                            | PSNR   | 19.051 | 23.901     | 19.369 | 24.509 | 21.708 |
> >
> > **R3-Q6: Ablation studies of depth mutual learning on other datasets**
> >
> > Thank the reviewer for the advice. Following the suggestion, we conduct the ablation studies of depth mutual learning on other datasets. The results are consistent with those presented in the main context of the paper, further proving the effect of depth mutual learning.
> >
> > | Method   | 360v2  | Free Dataset | ScanNet |
> > |----------|:------:|:------------:|:-------:|
> > | w/o dml  | 27.385 | 26.443       | 28.776  |
> > | GML-NeRF | **27.871** | **26.449**       | **28.805**  |
> >
> > **R3-Q7: Results of Switch-NGP with soft gate**
> >
> > Actually, the point-level fusion in Table 3 is a combined version of Switch-NGP and soft gate module in GML-NeRF, in which the model uses all the sub-NeRF outputs weighted by gating scores and completes the fusion process in the point dimension. The results are shown below.
> >
> > | Fusion-Dimension | PSNR   | SSIM  | LPIPS |
> > |------------------|:------:|:-----:|:-----:|
> > | Point-Level      | 20.796 | 0.661 | 0.413 |
> > | Ray-Level(Ours)  | **21.708** | **0.672** | **0.399** |

---

> ### Author Response · Authors · 2023-11-21
> **Follow-up**
>
> Dear Reviewer 3Q2w,
>
> Thanks once again for your valuable time in reviewing our paper. Did our previous responses address your concerns satisfactorily? If you have any further questions, please kindly inform us. We are more than willing to provide further clarifications.
>
> Authors

---

> ### Author Response · Authors · 2023-11-23
> **Looking forward to the reply**
>
> Dear reviewer 3Q2w,
>
> Thank you once more for the review. We eagerly anticipate your feedback on our response. Did our previous response and revision address all your concerns? If you have any further questions, please let us know and we will try our best to provide additional clarification before the DDL.
>
> Authors

---

> > ### Comment · Reviewer_3Q2w · 2023-11-23
> > **Thanks for the new results.**
> >
> > Thanks for the detailed response from the authors. They addressed most of my questions. The new experiments give some support to this method and meanwhile show some marginal or worse results. For example, the method w/o dml on 360v2 (PSNR 27.385) is worse than Block-NGP (PSNR 27.436) in Paper Table 1. Therefore, this result does not support the advantage of ray-based gating.

---

> > > ### Author Response · Authors · 2023-11-23
> > > **The comparison with Block-NGP**
> > >
> > > Thank the reviewer for the feedback.
> > >
> > > On other datasets, GML-NeRF (learnable ray-based allocation) shows better performance than Block-NGP (clustered view-based allocation) as shown in R3-Q6. Block-NGP is only comparable to GML-NeRF on the 360v2 dataset with a margin performance difference. **We add SSIM and LPIPS metrics below.** We guess that this is because the capture trajectory is regular and simple in 360v2 scenes. Besides, all views are towards the central object, and thus, only considering the image positions can also achieve a reasonable allocation. Indeed, compared with existing specially designed allocation mechanisms (e.g., block-nerf, mega-nerf), GML-NeRF may not be better on scenes with regular trajectory or special structure. However, the learnable ray-based allocation in GML-NeRF is more generally applicable and can achieve better performance on complex scenes.
> > >
> > > | Method     | PSNR  | SSIM  | LPIPS  |
> > > |----------|:------:|:------:|:------:|
> > > | Ours w/o DML   |27.385 |**0.771** |**0.288** |
> > > | GML-NeRF    | **27.436** | 0.761 | 0.298 |

---

### Official Review · Reviewer_PcY6 · 2023-11-04

**Soundness:** 3 good
**Presentation:** 2 fair
**Contribution:** 2 fair
**Rating:** 5
**Confidence:** 3

**Summary:**

This paper proposed two techniques to improve NeRF's modeling capability of complex scenes.
The paper jointly trains K NeRFs along with a learnable weighting scheme to combine the K NeRF's outputs into one.
The paper also proposes to use one NeRF's depth rendering to supervise another NeRF's depth output.
Experiments show that the proposed method outperforms many efficient NeRF baselines including DVGO, Instant-NGP.
However, the proposed method performs slightly worse than MipNeRF360.
The author argues that MipNeRF360 trains slowly, and also compares against MipNeRF360 trained only for one hour (which is about the same time needed by author's model to train). In this case, the proposed method works better.

**Strengths:**

The overall idea seems valid.
The paper is relatively easy to follow.

**Weaknesses:**

1. By the time of submission, I believe ZipNeRF paper was already out there and there was an open-source implementation of it. Gaussian Splatting was also available. Probably the author should compare against these two approaches too. For the following reasons: (a) ZipNeRF is built on Instant-NGP and is able to handle varying resolution training images which is more suitable than Instant-NGP to model relatively large (and unbounded) scenes; (b) Gaussian Splatting, by design, is able to handle the occlusion issue (mentioned by the author as a challenge in modeling complex scenes) as it's an explicit representation.

2. I'm a bit skeptical about the claim in experiments that since MipNeRF360 is too slow, you compare against MipNeRF360 trained for one hour. Is the main claim of the paper about efficiency? I mean, you can use always bump up the number of GPUs in training to reduce the training time if that's a problem.

3. It seems, by the end of the day, the author only uses two NeRFs (K=2) in their framework, though in the abstract the author claimed that they construct "an ensemble of sub-NeRFs". Does the method only work with K=2? If it works in more general settings with K > 2, the author should demonstrate that. Does having more sub-NeRFs help improve the rendering quality?

4. In Figure 5, the blue curve scales the width of a NeRF. Which NeRF's width is scaled? Is it any of the baseline NeRFs shown in Table 1? And on what dataset the experiment is conducted? Is the PSNR performance averaged across one scene, multiple scenes?

5. Also in Figure 5, I think it makes more sense to compare against the following: You scale the resolution of the feature grid used in Instant-NGP, and show how the performance of Instant-NGP scales. Since I think the key to model large scale scenes well is to leverage locality, and in Instant-NGP, you need to bump up the feature grid's resolution to better leverage locality.

**Questions:**

Figure 5. is a bit confusing. Why "GML-NeRF with two independent sub-NeRFs" does not overlap with the "2x NeRFs"?
I guess these are two different approaches? But I'm really confused here. The author could do a better job to clarify this.

Eq. (3), G(r)'s output is a K-dimensional vector? Do the elements of the K-vector sum to one or not? I guess not, as there is no such constraint imposed through losses.

---

> ### Author Response · Authors · 2023-11-19
> **Response to Reviewer PcY6-1**
>
> We thank the reviewer for the constructive feedback and thoughtful comments. We address the detailed questions and comments below.
>
> **R2-Q1: Results of ZipNeRF and Gaussian Splatting**
>
> Thank the reviewer for the advice regarding the comparison of GML-NeRF with the latest baselines.
>
> (1) We implement a ZipNeRF version of GML-NeRF, named GML-ZipNeRF, and evaluate the performance on the 360v2 dataset. The training settings are kept the same as the original paper, including the training iterations and batch size. We add the comparison results in the Appendix G, showing that integrated with GML-NeRF, ZipNeRF can also obtain performance gains.
>
> (2) Due to time constraints, we could not verify the effect of GML-NeRF upon the Gaussian Splatting. A possible implementation to integrate GML-NeRF and 3D Gaussian Splatting is constructing an ensemble of 3D Gaussian representations for independent rendering. Subsequent gate-guided fusion can be conducted in a similar way to this article. We leave it for future exploration.
>
> | Method      | Metric | bicycle | bonsai | counter | garden | kitchen | room   | stump  | avg    |
> |-------------|:------:|:-------:|:------:|:-------:|:------:|:-------:|:------:|:------:|:------:|
> | ZipNeRF     | PSNR   | 21.019  | 33.052 | 25.982  | 24.33  | 32.843  | 34.777 | 25.406 | 28.201 |
> |             | SSIM   | 0.616   | 0.962  | 0.852   | 0.758  | 0.971   | 0.951  | 0.722  | 0.833  |
> | GML-ZipNeRF | PSNR   | 20.488  | 33.486 | 26.372  | 24.603 | 33.1197 | 35.795 | 25.581 | **28.492** |
> |             | SSIM   | 0.601   | 0.965  | 0.859   | 0.765  | 0.9731  | 0.956  | 0.728  | **0.835**  |
>
> As a multi-NeRF training framework, GML-NeRF is essentially orthogonal to the structure and training method of single-NeRF. In our work, for the benefit of training efficiency and its wide application, we build and validate GML-NeRF upon the Instant-NGP. Nevertheless, it can be integrated with other single-NeRF frameworks, including ZipNeRF and Gaussian Splatting, and so on.
>
> **R2-Q2: Consideration of training efficiency**
>
> Thank the reviewer for the important reminder. Considering the importance of training efficiency on the practicability of the NeRF training frameworks, we mainly choose the baselines with high training efficiency (NGP, DVGO, F2-NeRF, etc.) and other multi-NeRF training methods for comparison, while providing the results of other baselines as the extra reference (NeRF, MipNeRF360,...). The consideration for training efficiency is also the motivation for adopting hybrid representations in GML-NeRF. Using the hybrid representation, ensemble size growth only increases the number of MLPs, which brings only a moderate parameter size increase. We highlight this consideration in the structure design part of the method section in the revision.
>
> Additionally, although our GML-NeRF implemented upon Instant-NGP may not exhibit comparable performance to MipNeRF360 trained with more than one day, it acts as a flexible multi-NeRF training framework with the potential to be integrated with various single-NeRF frameworks (NeRF, MipNeRF, etc). As the above experimental results show, implemented upon SOTA single-NeRF method ZipNeRF, an NGP-version of MipNeRF360, GML-ZipNeRF shows better rendering quality, which further proves the effectiveness of the proposed method.
>
> **R2-Q3: Results of more sub-NeRFs**
>
> We adopt the k=2 setting by default mainly for the sake of experiment convenience, given its lower memory footprint in training. Nevertheless, GML-NeRF can consistently obtain performance gains as the number of sub-NeRFs increases, as shown in Figure-5.
>
> To provide a more comprehensive demonstration of this point, we provide per-scene results of GML-NeRF with more sub-NeRFs on the ScanNet dataset in the Appendix F. As the results show, GML-NeRF with 4x sub-NeRFs performs better than the one with 2x sub-NeRFs across all scenes.
>
> | Method    | Metric | 0046_00 | 0276_00 | 0515_00 | 0673_04 | Avg    |
> |---------|:------:|:-------:|:-------:|:-------:|:-------:|:------:|
> | GML-size2 | PSNR   | 29.440  | 30.871  | 29.149  | 25.759  | 28.805 |
> | GML-size3 | PSNR   | 29.878  | 31.242  | 29.47   | 25.944  | 29.134 |
> | GML-size4 | PSNR   | 30.018  | 31.31   | 29.679  | 26.063  | **29.268** |

---

> ### Author Response · Authors · 2023-11-19
> **Response to Reviewer PcY6-2**
>
> **R2-Q4: Demonstration of scalability study and the responding figure**
>
> Thank the reviewer for pointing out several issues and questions regarding Figure 5. The feedback is important for refining the paper, and we would like to address the concerns below.
>
> (1) Scalability results clarification: We acknowledge the omission that we only report the scalability results on scene 0046 of the ScanNet dataset rather than the average PSNR result of four scenes. We have modified it in the revision.
>
> (2) Hidden dimension scaling for the blue curve: The blue curve scales the hidden dim of both the geometry MLP and the color MLP while maintaining the resolution of the hash feature grid.
>
> (3) GML-NeRF with two independent sub-NeRFs: This implementation is different from the one of GML-NeRF, in which the difference is whether the feature grid is shared among the sub-NeRFs. The number of parameters increases significantly within the design of unshared feature grids. We have slightly modified this figure in the revision to clarify this detail.
>
> We hope that our modification and clarification can help!
>
> **R2-Q5: Effect of scaling up the resolution of feature grid**
>
> Following the reviewer's suggestion, we further conduct experiments to explore the impact of increasing the number of parameters in the feature grid, and the results are incorporated into Figure-5 in the latest revision. While maintaining default values for resolution and the number of hash levels, we opted to increase the hash table size to scale up the feature grid, consistent with the test methodology employed in the original NGP paper. As the results show, although the number of parameters increases significantly, the performance gains achieved by increasing the "log2_hash_table_size" from 19 to 24 are limited, which is consistent with the results in the original paper.
>
> | Method    | 0046_00 | 0276_00 | 0515_00 | 0673_04 | Avg    | Params/MB |
> |-----------|:-------:|:-------:|:-------:|:-------:|:------:|:---------:|
> | NGP(t=19) | 28.71   | 30.08   | 28.182  | 25.324  | 28.074 | 24.960    |
> | NGP(t=21) | 28.888  | 30.243  | 28.405  | 25.593  | 28.282 | 91.303    |
> | NGP(t=24) | 29.133  | 30.459  | 28.569  | 25.753  | 28.478 | 615.59    |
> | GML-NeRF  | **29.581**  | **30.884**  | **29.172**  | **25.844**  | **28.870** | **24.981**    |
>
> **R2-Q6: G(r) in Equ-3**
>
> G(r) is the K-dimensional vector representing the prediction confidence of each sub-NeRF for the ray r. Since we insert a softmax normalization layer at the end of the gate module, the elements of the G(r) sum to one.

---

> ### Author Response · Authors · 2023-11-21
> **Follow-up**
>
> Dear Reviewer PcY6,
>
> Thanks once again for your valuable time in reviewing our paper. Did our previous responses address your concerns satisfactorily? If you have any further questions, please kindly inform us. We are more than willing to provide further clarifications.
>
> Authors

---

> ### Author Response · Authors · 2023-11-23
> **Looking forward to the reply**
>
> Dear reviewer PcY6,
>
> Thank you once more for the review. We eagerly anticipate your feedback on our response. Did our previous response and revision address all your concerns? If you have any further questions, please let us know and we will try our best to provide additional clarification before the DDL.
>
> Authors

---

### Official Review · Reviewer_MyQ8 · 2023-11-07

**Soundness:** 2 fair
**Presentation:** 3 good
**Contribution:** 2 fair
**Rating:** 5
**Confidence:** 5

**Summary:**

The paper introduces a new learnable multi-NeRF framework for neural rendering, adopting partitioning and using different models to capture a scene.  The method aims to address the point visibility problem in Switch-NeRF by assigning confidence score with respect to rays rather than 3D points. The work is also different from other ray/pixel-based approaches in which allocation rule is manually-defined. A depth loss regularization is used to guide the training of subnets besides the default color supervision. Experiments on multiple datasets shows that the proposed method can achieve competitive rendering results quantitatively and qualitatively.

**Strengths:**

1. Using scene partitioning and learning with multiple subnetworks is a promising direction for improving the scalability and performance of scene rendering tasks. The authors propose a simple and effective learnable framework and have shown promising experimental results.

2. The paper is well-organized and easy to follow. The authors also present sufficient discussion for comparing with related work.

3.  Experiments on multiple-datasets and the comparison with modern SOTA methods are provided to verify the method and the modules inside.

**Weaknesses:**

1. I have some concerns regarding the validity of the depth regularization. The fused depth is estimated by weighting depths from different subnets (which is named as sub-depth for simplification). In this regard, the sub-depth with higher weighting score is supposed to be closer to the target depth, thus the difference between them should be smaller, and the one between the sub-depth with lower weighting score and the target is large.  Simply summing them together may cause the training penalize more for the sub-depth with lower weighing score, which seems less proper.  The authors need to give more discussion (may with ablation study) about it.

2. The ablation study with more subnetworks can achieve better rendering result while a smaller value (i.e., 2) is used by default in this paper. What’s the consideration behind it? I think using a larger number of subnets may introduce difficulty for training. The difficulty in training convergence may significant grow as ray capacity and the number of subnetwork increases, as there may be a great number of possible arrangements. In contrast, using prior knowledge (e.g., nearby points or pixels may highly go through the same subnet) is likely to be helpful for reducing the complexity. The authors need to give more discussion about it as well as analysis on training complexity.

3. For a detail in the design, is sharing the feature grid necessary?  The authors mentioned that it is helpful for reducing parameters. If using different feature grids for subnets, it may introduce some problems due to the lack of global index.  It would be better to give some discussion and experimental result to validate the design.

4. The authors need to discuss the limitation of the work. It is unclear that if the method can handle the scene with translucent objects. The method is built on the assumption that each ray is associated with a single confidence score, i.e., depth value. But depth estimation is naturally a problem for translucent objects.

**Questions:**

Please see the questions in the ``Weaknesses".

---

> ### Author Response · Authors · 2023-11-19
> **Response to Reviewer MyQ8-1**
>
> We thank the reviewer for the constructive feedback and thoughtful comments. We address the detailed questions and comments below.
>
> **R1-Q1: Mechanism of depth regularization**
>
> Thank the reviewer for the questions. To empirically analyze the effect of depth mutual learning, we conduct an ablation study on the TAT dataset and add the results to the Appendix.D following the reviewer's suggestion. When using the average of the sub-depths as the target depth (all sub-NeRFs have equal regularization strength), the rendering quality will be slightly worse (PSNR 21.419 vs 21.708). This observation highlights the pivotal role of gate-guided depth mutual learning. In the GML-NeRF framework, an ensemble of sub-NeRFs is jointly optimized via the soft gate module, a crucial component enhancing rendering consistency. Within the design of the soft gate module, all the sub-NeRFs contribute to the same ray, necessitating accurate encoding of scene geometry. Meanwhile, the sub-NeRF with a lower gating score exhibits a relatively poor specialty for the ray, resulting in the inaccuracy of geometry encoding. A larger regularization force could have a better effect in such cases. By differently penalizing sub-depths based on their respective gating scores, our approach significantly contributes to accurate scene representation and improved rendering quality.
>
> | Method | Metric | M60 | Playground | Train | Truck | Avg |
> |:---------------:|:---------------:|:------------:|:-------------------:|:--------------:|:--------------:|:------------:|
> | Equal DML       | PSNR            | 18.929       | 23.108              | 19.012         | 24.625         | 21.419       |
> | Ours            | PSNR            | 19.051       | 23.901              | 19.369         | 24.509         | **21.708**       |
>
> **R1-Q2: Analysis of training complexity**
>
> Thank the reviewer for the thoughtful questions. For the convenience of the experiments (lower memory footprint in training), we adopt the k=2 setting by default. However, it is crucial to note that GML-NeRF consistently obtains performance gains as the number of sub-NeRFs increases, as shown in Figure-5. We add per-scene results in the Appendix F for reference.
> Moreover, **the training difficulty will not increase significantly with the increase in the number of sub-NeRFs**. In our experiments, we find that the model with four sub-NeRFs converges faster than the one with two sub-NeRFs while achieving better rendering quality with the same training iterations. The convergence curves are also added to the Appendix F. The ease of training convergence can be attributed to two aspects. On the one hand, the feature grid is shared among multi-NeRFs, and thus, the number of learnable parameters increases marginally. On the other hand, as the neural network is better at fitting low-frequency information, our gate module (a 4-layer MLP without sinusoidal position encoding) has implicitly incorporated "smoothness prior", leading to closer rays to be more possibly assigned closer gating scores. Such smoothness prior is similar to what the reviewer mentioned, helping to reduce the training complexity with more sub-NeRFs.
>
> | Method | Metric | 0046_00 | 0276_00 | 0515_00 | 0673_04 | Avg |
> | --- | --- | --- | --- | --- | --- | --- |
> | GML-size2 | PSNR | 29.440 | 30.871 | 29.149 | 25.759 | 28.805  |
> | GML-size3 | PSNR | 29.878 | 31.242 | 29.470 | 25.944 | 29.134  |
> | GML-size4 | PSNR | 30.018 | 31.310 | 29.679 | 26.063 | **29.268**  |

---

> ### Author Response · Authors · 2023-11-19
> **Response to Reviewer MyQ8-2**
>
> **R1-Q3: Effect of shared feature grid**
>
> Thank the reviewer for the questions. In GML-NeRF, the shared feature grid is a better choice than the unshared one. We conduct an ablation study on the TAT dataset and add the results to the Appendix D. The experiment results show that the model employing a shared feature grid outperforms its counterpart with multiple independent feature grids. Such superiority comes from two aspects.
>
> Firstly, within the hybrid representation, the feature grid is responsible for encoding features of 3D spatial points, while the MLP encoder is designed to encode view information. The crucial design of independent MLP decoders aligns with our visibility-aware motivation, thereby enhancing the view-dependent effect. Secondly, the training complexity will also increase as the trainable parameters increase. With the limited amount of training data, increasing the number of feature grids leads to poor convergence. By contrast, as different rays may pass through the same region of 3D space, weight sharing for the feature grid helps to facilitate training.
>
> Although the model with unshared feature grids can still achieve satisfactory results, particularly in terms of the SSIM and LPIPS metrics, we still recommend using the shared feature grid scheme due to its parameter efficiency.
>
> | Method                  | Metric | M60    | Playground | Train  | Truck  | Avg    |
> |-------------------------|:------:|:------:|:----------:|:------:|:------:|:------:|
> | unshared\_feature\_grid | PSNR   | 18.765 | 22.839     | 18.958 | 24.493 | 21.264 |
> | Ours-size2              | PSNR   | 19.051 | 23.901     | 19.369 | 24.509 | 21.708 |
>
> **R1-Q4: Limitation of GML-NeRF on translucent objects**
>
> Thank the reviewer for the insightful suggestion.
>
> We think one limitation of GML-NeRF is that compared with single-NeRF or multi-NeRF (with hard gating) methods, it will introduce additional inference latency, although achieving better rendering performance. When the ensemble size is large, we might need to choose several sub-NeRFs instead of using all sub-NeRFs to reduce the inference latency. In addition, in order to strike a good balance between the rendering performance and inference latency, we might need to investigate how to decide the best choice of ensemble size (i.e., $K$) for a given scene.
>
> As for the handling of translucent objects. In our view, we think the difficulty of tackling is a problem faced by all NeRF methods, and GML-NeRF doesn't bring extra difficulty. We'd like to know if the reviewer thinks that GML-NeRF has introduced extra in rendering transparent objects than other work and look forward to further suggestions.

---

> ### Author Response · Authors · 2023-11-21
> **Follow-up**
>
> Dear Reviewer MyQ8,
>
> Thanks once again for your valuable time in reviewing our paper. Did our previous responses address your concerns satisfactorily? If you have any further questions, please kindly inform us. We are more than willing to provide further clarifications.
>
> Authors

---

> ### Author Response · Authors · 2023-11-23
> **Looking forward to the reply**
>
> Dear reviewer MyQ8,
>
> Thank you once more for the review. We eagerly anticipate your feedback on our response. Did our previous response and revision address all your concerns? If you have any further questions, please let us know and we will try our best to provide additional clarification before the DDL.
>
> Authors

---

### Author Response · Authors · 2023-11-19
**Paper changes**

Dear All,

We appreciate all reviewers' valuable time invested in reviewing our paper. We are encouraged that the reviewers recognize the neat idea (MyQ8,PcY6,3Q2w,Vfdc), extensive experiments(MyQ8,3Q2w), insightful motivation(3Q2w), flexible and effective for practical adoption (MyQ8,PcY6,Vfdc), and good presentation (MyQ8, PcY6). We are also thankful for all the concerns and suggestions.

According to the suggestions, we have revised the paper as belows:
* Add the consideration of training efficiency in the structural design of Section 4.2.
* Add the advantages of shared feature grid in the structural design of Section 4.2.
* Highlight the mechanism of CV balanced regularization in Section 4.4.
* Modify Figure-5 (add new results with scaling up the feature grid) and the related words for clarification.
* Add the reference for ZipNeRF.
* Add more ablation studies on GML-NeRF's components in Appendix.D.
* Add the discussion and experiments of Mega-NGP in Appendix.E.
* Add more scalability results such as convergence curves in Appendix.F.
* Add the discussion and experiments of GML-ZipNeRF in Appendix.G.

---

### Meta-Review · Area_Chair_4HV8 · 2023-12-13

**Metareview:**

This paper receives 4x marginally below the acceptance threshold. The major weaknesses of the paper include: concerns regarding the validity of the depth regularization. Lack of comparison against ZipNerf and Guassian Splatting. Doubts over experimental settings for comparison against MipNerf360 and Instant-NGP scales. Unclear on why the proposed method gets very good quality improvement with a minor increase of parameters. It is also unclear how the Image-level fusion is performed.  The addition result shown in the rebuttal does not support the advantage of ray-based gating. The soft gate module's mechanism is not comprehensively explored. There appears to be a discrepancy between the motivating concept presented in Figure 1 (b) and the ray allocations depicted in Figure 4 (left), where rays are equally distributed between two NeRFs (i.e., MLPs).

**Justification For Why Not Higher Score:**

There are doubts on the proposed method and experimental results.

**Justification For Why Not Lower Score:**

N/A

---

### Decision · Program_Chairs · 2024-01-16

Reject